# MrT5: Dynamic Token Merging for Efficient Byte-level Language Models

**Julie Kallini, Shikhar Murty, Christopher D. Manning, Christopher Potts, Róbert Csordás**
Stanford University
{kallini, jsmurty, manning, cgpotts, rcsordas}@stanford.edu

## ABSTRACT

Models that rely on subword tokenization have significant drawbacks, such as sensitivity to character-level noise like spelling errors and inconsistent compression rates across different languages and scripts. While character- or byte-level models like ByT5 attempt to address these concerns, they have not gained widespread adoption—processing raw byte streams without tokenization results in significantly longer sequence lengths, making training and inference inefficient. This work introduces **MrT5** (**Me**rge**T5**), a more efficient variant of ByT5 that integrates a token deletion mechanism in its encoder to *dynamically* shorten the input sequence length. After processing through a fixed number of encoder layers, a learned *delete gate* determines which tokens are to be removed and which are to be retained for subsequent layers. MrT5 effectively "merges" critical information from deleted tokens into a more compact sequence, leveraging contextual information from the remaining tokens. In continued pre-training experiments, we find that MrT5 can achieve significant gains in inference runtime with minimal effect on performance, as measured by bits-per-byte. Additionally, with multilingual training, MrT5 adapts to the orthographic characteristics of each language, learning language-specific compression rates. Furthermore, MrT5 shows comparable accuracy to ByT5 on downstream evaluations such as XNLI, TyDi QA, and character-level tasks while reducing sequence lengths by up to 75%. Our approach presents a solution to the practical limitations of existing byte-level models.

 https://github.com/jkallini/mrt5

## 1 INTRODUCTION

*Subword tokenization*, typically via algorithms such as byte-pair encoding (Sennrich et al., 2016) or SentencePiece (Kudo & Richardson, 2018), is a fundamental text preprocessing step that has become ubiquitous in modern large language models (LLMs). Subword tokenizers divide text into meaningful units known as *tokens*, which closely resemble words or parts of words. Tokenization can be seen as a form of compression, since it reduces the sequence length of the input passed to the compute-intensive Transformer (Vaswani et al., 2017). However, subword tokenizers have several drawbacks. For example, they are not very robust to character-level noise and manipulations, such as spelling errors (Kaushal & Mahowald, 2022; Huang et al., 2023); they directly impact how models process digits and perform arithmetic (Singh & Strouse, 2024; Zhou et al., 2024); and they have disproportionate compression rates for different languages and scripts (Ahia et al., 2023; Petrov et al., 2023). In addition, current language model APIs charge users per-token, and such discrepancies can cause users of certain languages to be overcharged due to poorer compression.[1]

As an alternative to subword models, *tokenization-free* models skip the tokenization preprocessing step entirely by passing the raw character or byte stream directly as input. However, character- or byte-level sequences tend to be significantly longer than tokenized text sequences, which limits the practical utility of tokenization-free models. For example, ByT5 (Xue et al., 2022), a byte-level counterpart of mT5 (Xue et al., 2021), is competitive with mT5 on a number of tasks, but it has a much slower pre-training and inference runtime. Most other tokenization-free models explicitly

---

[1]See also Andrej Karpathy's tweets on tokenization: https://x.com/karpathy/status/1759996551378940395; https://x.com/karpathy/status/1657949234535211009

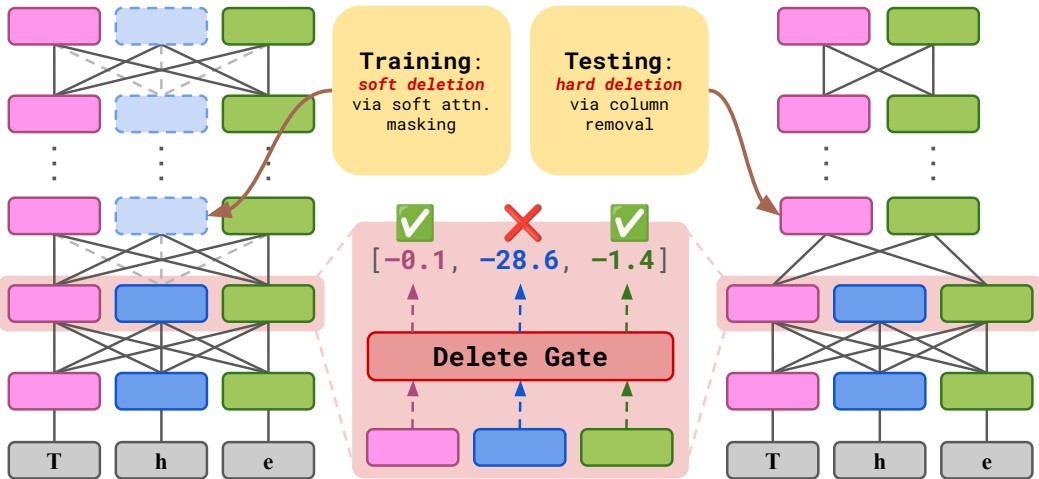

Figure 1: MrT5's encoder during training and testing. During training, fully-differentiable *soft deletion* masks out tokens using the output of MrT5's delete gate. During testing, *hard deletion* removes columns from the computation, which reduces the sequence length and leads to efficiency gains. In this visual, the delete gate is placed at layer 2, but the gate placement may be tuned.

downsample or pool representations to reduce the sequence length (Clark et al., 2022; Tay et al., 2022; Nawrot et al., 2022; 2023). Many of these architectures rely on fixed-span downsampling methods; however, meaningful units of text often span a variable number of bytes or characters. While some models identify variable-length spans, they introduce substantial modifications to the standard Transformer architecture, making it difficult to adapt existing pre-trained models.

In this work, we explore the question: can we make an existing byte-level model more efficient? We propose **MrT5** (**M**erge**T5**), a variant of the ByT5 architecture that addresses its inefficiencies while maintaining its performance (Section 3). MrT5 dynamically merges its encoder's input into a shorter sequence using a token deletion gating mechanism at a fixed, early encoder layer, as shown in Figure 1. By allowing the first few encoder layers to process the entire sequence, the encoder creates contextualized representations of the tokens. When the gating mechanism then deletes a subset of the tokens, those that remain keep the contextual information about those that were removed, allowing information to be implicitly merged into a shorter sequence. During training, we use a deletion regularizer with a tunable weight that can adjust the amount of deletion MrT5 performs. MrT5 effectively learns to merge relevant tokens and delete extraneous ones in a completely unsupervised manner, by optimizing the balance between the regularization objective and language modeling.

We first train several MrT5 models on diagnostic tasks (Section 4) and find that MrT5 not only drops irrelevant tokens but also compresses relevant context using task-specific deletion patterns. Next, we perform continued pre-training experiments by fine-tuning the MrT5 gating mechanism on top of a pre-trained ByT5 (Section 5). Our results show that MrT5 achieves lower bits-per-byte than both random and fixed token deletion baselines, as well as pooling-based alternatives, at the same compression rates. With multilingual training, MrT5 adjusts to each language's orthographic features, learning optimal compression rates specific to each language. Finally, in multilingual and character-level benchmarks (Section 6), MrT5 achieves comparable accuracy to ByT5 while cutting the sequence length by up to 75%, significantly improving inference runtimes. Our approach improves on the main limitations of ByT5, representing a significant step toward the adoption of byte-level language models and the elimination of subword tokenization from modern NLP.

## 2 BACKGROUND: TOKENIZATION-FREE MODELS

**Soft Tokenization and Downsampling Methods.** Many character- or byte-level models employ "soft tokenization" or explicit downsampling to shorten input sequences. Here, we focus on specialized Transformer architectures. CANINE (Clark et al., 2022), a character-level counterpart to mBERT (Devlin et al., 2019), uses convolutional downsampling before feeding inputs to a 12-layer

Transformer encoder. Charformer (Tay et al., 2022) learns a gradient-based block scoring function to pool byte embeddings for efficient training and inference. Islam et al. (2022) propose a vocabulary-free neural tokenizer trained via supervision from a heuristic-based subword tokenizer.

MegaByte (Yu et al., 2023) scales byte-level decoders to long-context tasks by segmenting sequences into fixed-length "patches," though these may not align with meaningful units of text. SpaceByte (Slagle, 2024) applies global Transformer blocks to specific byte types, such as spaces, but these bytes are also not chosen dynamically. Hourglass Transformers (Nawrot et al., 2022) incorporate hierarchical downsampling and upsampling at different layers within decoder models, and the architecture was later extended with a gradient-based tokenization module that dynamically pools characters using a boundary predictor (Nawrot et al., 2023; Ahia et al., 2024). MANTa (Anagnostidis et al., 2024) also learns token boundaries via sliding window attention. More recently, the Byte Latent Transformer (BLT, Pagnoni et al., 2024) and EvaByte (Zheng et al., 2025) have shown that tokenizer-free models can scale more efficiently than subword LLMs. BLT utilizes a dynamic boundary predictor based on byte entropies to segment sequences into variably sized patches, and EvaByte uses multibyte prediction and linear attention to improve scalability and decoding speed.

**ByT5.** This paper focuses on ByT5 (Xue et al., 2022), a byte-level sequence-to-sequence Transformer architecture designed as a counterpart to mT5 (Xue et al., 2021), the multilingual extension of T5 (Raffel et al., 2020). Unlike the models discussed previously, ByT5 uses no soft tokenization or downsampling steps to reduce the sequence length. To compensate for the loss of the parameters that would normally be used for subword embeddings, ByT5 has a "heavy" encoder with more layers than the decoder. While ByT5 matches or outperforms mT5 on a variety of downstream tasks, its heavy encoder, large model and feed forward dimensionalities, and short input sequence length (1024 bytes) make it quite inefficient. ByT5 requires 1.2 times more operations than mT5, and it can be up to 10 times slower at inference, depending on the encoder's input length.

MrT5 modifies the ByT5 architecture to make it more efficient. Unlike previous work, MrT5's deletion gating mechanism can be applied to a pre-trained model with minimal fine-tuning and few additional parameters, or to models trained from scratch. For further related work on adjacent topics like early exit models, long-context, and additional tokenization-free architectures, see Appendix A.

## 3  THE MRT5 MODEL ARCHITECTURE

MrT5 introduces a unique deletion gating mechanism: in a fixed encoder layer, a *delete gate* selects which tokens to keep for further processing and which to discard, effectively merging information into a shorter sequence. We apply this gating at a single layer for three reasons: (1) to avoid the overhead of executing the deletion algorithm multiple times, (2) because performance stabilizes after early layers (Section 7), and (3) because early deletion yields the greatest computational savings.

### 3.1  DELETION GATING MECHANISM

The MrT5 deletion gating mechanism is inspired by existing architectures with gating mechanisms such as Long-Short Term Memory (LSTMs, Hochreiter & Schmidhuber, 1997), Gated Recurrent Units (GRUs, Cho et al., 2014), and Mixture-of-Experts (MoEs, Shazeer et al., 2017). MrT5's delete gate is placed after the output of a fixed encoder layer $l$ and is defined by the following function:

$$\mathbf{G} = k\sigma(\text{LayerNorm}(\mathbf{H}_l)\mathbf{W} + \mathbf{1}_N b) \tag{1}$$

where $\mathbf{H}_l \in \mathbb{R}^{N \times d_{\text{model}}}$ are the hidden states output by layer $l$; $\mathbf{W} \in \mathbb{R}^{d_{\text{model}} \times 1}$; $b \in \mathbb{R}$; $\mathbf{G} \in \mathbb{R}^{N \times 1}$; $k$ is a large negative constant; $N$ is the encoder input sequence length; $d_{\text{model}}$ is ByT5's hidden state/model dimensionality; and $\mathbf{1}_N \in \mathbb{R}^{N \times 1}$ is a vector of ones. $\text{LayerNorm}(\mathbf{H}_l)$ denotes the application of layer normalization. Following the T5 architecture, this is implemented as *root mean square* layer normalization (RMSNorm, Zhang & Sennrich, 2019). The gating activation function is a rescaled and translated sigmoid function, bounded between $k$ and 0. In our experiments, we use $k = -30$. During training, our experiments include adding Gumbel noise to the gate's logits to encourage exploration of gate values. However, we found that the model performs well even without it, making this step optional. See Appendix B for the Gumbel noise formulas.

MrT5's delete gate introduces only $2d_{\text{model}} + 1$ additional parameters in total; $d_{\text{model}}$ for the weight vector $\mathbf{W}$, $d_{\text{model}}$ for the layer normalization, and one for the bias term. This makes the method highly parameter-efficient.

**Soft and Hard Deletion.** During training, MrT5 deletes tokens *softly*, where the outputs of the gating mechanism $\mathbf{G}$ are applied as a soft attention mask. These outputs are added directly to the self-attention mechanism of the subsequent encoder layers, as well as to the cross-attention layers between the decoder and encoder. The modified attention mechanism for each attention head is defined as:

$$\text{SoftDeletionAttention}(\mathbf{Q}, \mathbf{K}, \mathbf{V}) = \text{softmax}\left(\frac{\mathbf{Q}\mathbf{K}^\top}{\sqrt{d_{\text{head}}}} + \mathbf{1}_N\mathbf{G}^\top\right)\mathbf{V} \tag{2}$$

where $d_{\text{head}}$ is the dimensionality of the query, key, and value vectors for a single attention head; $\mathbf{Q}, \mathbf{K}, \mathbf{V} \in \mathbb{R}^{N \times d_{\text{head}}}$; and $\mathbf{1}_N \in \mathbb{R}^{N \times 1}$ is a vector of ones. A token at sequence position $i \in [1, N]$ with $\mathbf{G}_i \approx 0$ will not be masked, whereas a token at position $j \neq i \in [1, N]$ with $\mathbf{G}_j \approx k$ will be masked, since $k$ is a large negative constant. Though soft deletion does not reduce the sequence length, we apply it during training to emulate the effect of token deletion while being fully differentiable. To see efficiency gains during inference, we apply *hard deletion*, where the hidden states are removed from the sequence, determined by a hard threshold; we set this threshold to be $\frac{k}{2}$, half of the range of the delete gate's output.

In a batch, different samples may have different numbers of tokens deleted. With hard deletion, the new sequence length is set by the longest remaining sequence, and shorter ones are padded. Since T5 uses relative position biases within the attention mechanism of each layer, deletion and padding is performed on both the hidden states and position biases. For a theoretical analysis of MrT5's compute savings with hard deletion, see Appendix C.

## 3.2 GATE REGULARIZER

MrT5 allows deletion rates to be adjusted using a tunable regularizer loss:

$$\mathcal{L}_{\mathbf{G}} = \frac{1}{N}\sum_{i=1}^{N}\mathbf{G}_i \tag{3}$$

This loss is the average of the gate output values, which encourages them to be more negative (i.e. closer to $k$, the minimum gate value). In other words, as this loss decreases, the number of deleted tokens increases. The total loss is defined as the sum $\mathcal{L} = \mathcal{L}_{\text{CE}} + \alpha\mathcal{L}_{\mathbf{G}}$, where $\mathcal{L}_{\text{CE}}$ is the cross-entropy loss. Varying the hyperparameter $\alpha$ allows the MrT5 model to delete more or fewer tokens.

**Optimizing for a Specific Deletion Ratio.** Setting $\alpha$ by hand allows the model to dynamically discover the deletion ratio depending on the difficulty of the task. However, we can also optimize for a specific ratio of deleted tokens using an algorithm that resembles the proportional-integral controller (PI controller) of classical control theory. We additionally use an exponential moving average on the P-term. Let's call the target deletion ratio $\delta \in [0, 1]$, the proportion of deleted tokens in the current batch $\hat{\delta}_t \in [0, 1]$, and the regularization hyperparameter for the current batch $\alpha_t$. We update $\alpha$ as follows:

$$p_{t+1} = \gamma p_t + (1 - \gamma)(\delta - \hat{\delta}_t) \tag{4}$$

$$i_{t+1} = i_t + \delta - \hat{\delta}_t \tag{5}$$

$$\alpha_{t+1} = \text{clamp}\left(k_p p_{t+1} + k_i i_{t+1}\right) \tag{6}$$

where $\text{clamp}(x) = \max(x, 0)$. We initialize $p_0 = 0$ and $i_0 = 0$. In all of our experiments, we use $\gamma = 0.9$. For most experiments (unless otherwise noted), we found that using $k_p = 0.5$ and $k_i = 1\mathrm{e}{-5}$ worked well in practice. This method is easier to use than manually setting $\alpha$ and allows $\alpha$ to change dynamically as the model undergoes phase transitions during training, resulting in more stable learning. Besides our diagnostic task models, all MrT5 models are trained with this PI controller.

**Softmax$_1$.** It is possible for all elements of $\mathbf{G}$ to equal the minimum gate value such that $\mathbf{G}_i = k$ for all $i \in [1, N]$. This $\mathbf{G}$ would satisfy the gate regularizer but fail to act as an attention mask, since adding the same value to all elements of the input to a standard $\text{softmax}$ function does not affect its output. To help avoid this scenario, we use $\text{softmax}_1$ (Miller, 2023) in the attention mechanism:

$$(\text{softmax}_1(\mathbf{x}))_i = \frac{\exp(\mathbf{x}_i)}{1 + \sum_j \exp(\mathbf{x}_j)} \tag{7}$$

Table 1: Diagnostic tasks with example input and target sequences. In our experiments, sequences are 64 characters/tokens long, including a start and end token. Legend: vowels to remove, vowels to keep, sequences to replace.

| Task | Input | Target |
|---|---|---|
| Simple Vowel Removal | zEKRreJcBxGUJQbZSIos | zKRrJcBxGJQbZSs |
| Contextual Vowel Removal | EOubXgaYVbiOgiIrEnld | EOubXgYVbOgIrnld |
| Sequence Merge | KjAxIpABCZCxBcniABCs | KjAxIpDZCxBcniDs |

With the $\text{softmax}_1$ function, if $\mathbf{G}_i = k$ for all $i \in [1, N]$, as $k$ becomes negative, the sum of attention scores approaches zero. This eliminates the failure case of using tokens that appear to be all deleted. For consistency, we use $\text{softmax}_1$ in all attention modules for MrT5 and baseline ByT5 models.

When training from scratch, we also found that attention scores could inflate to counteract the masking effect of the delete gate; we found that a simple attention score regularizer mitigated this issue, as described in Appendix D.

## 4 SIMULATIONS

We first train tiny 15M-parameter MrT5 and T5 models with 3 encoder layers and 3 decoder layers from scratch on three diagnostic tasks: a simple vowel removal task, a contextual vowel removal task, and a sequence merge task. The purpose of these experiments is to verify that the architecture behaves as intended, particularly with regard to the merging patterns it learns. Does MrT5 merely drop tokens when they are irrelevant to the output, or does it effectively merge relevant context into a shorter sequence? These results help set the stage for our continued pre-training experiments in Section 5 and our downstream task evaluations in Section 6.

**Diagnostic Task Specifications.** Each of our three diagnostic tasks is a variant of a copy task, designed to assess MrT5's ability to identify unimportant or redundant information in the input sequence or merge relevant information from some tokens into other tokens. Input sequences are 64 bytes long and are comprised of random lowercase and uppercase English characters, plus start and end tokens. Example inputs and labels are provided in Table 1.

1. **Simple Vowel Removal**: Generate a copy of the input token sequence, except for any vowels. We expect MrT5 to delete vowels, which occur with 19% probability. Thus, the optimal sequence length decrease is 19%.

2. **Contextual Vowel Removal**: Generate a copy of the input token sequence, except for any vowels that follow a lowercase consonant. We increase the probability of vowels such that, on average, they comprise 40% of the sequence, and about 18% of the sequence is comprised of vowels that follow a lowercase consonant. If MrT5 learns the relevant deletion pattern, we would expect a sequence length decrease of 18%.

3. **Sequence Merge**: Generate a copy of the input token sequence, and translate any occurrence of the character sequence 'ABC' into the character 'D'. 'ABC' sequences are inserted into the input randomly and occur about 5 times per sequence on average. If MrT5 merges 'ABC' into a single token, we would expect it to drop 10 tokens on average, which is 15.6% of the sequence.

In these experiments, we do not use a controller to enforce a specific deletion ratio. Instead, we train multiple models with different fixed $\alpha$ values to observe the various deletion patterns that naturally emerge. In all models, we place the delete gate at the middle encoder layer $l = 2$. For further model architecture and training configurations we use for the simulations, see Appendix E.1.

**Results.** Table 2 presents the performance of several MrT5 models that use different regularizer $\alpha$ values trained on each of our three diagnostic tasks. In all three diagnostic tasks, MrT5 models learned to selectively drop tokens in a way that suited each task's specific requirements. This allowed them to achieve the same performance as T5 models, while processing shorter sequences. For instance, in the simple vowel removal task, the MrT5 models selectively dropped vowels but kept consonants intact. These findings suggest that MrT5 can create complex deletion strategies that exploit patterns or redundancies in the input.

Table 2: Diagnostic task performance for T5 and MrT5 with different deletion strategies. Token-level accuracy measures the percentage of correctly predicted tokens, averaged across sequences; sequence-level accuracy measures the percentage of sequences with all tokens predicted correctly. MrT5 learns to delete around the optimal rate for each task while maintaining performance.

| Task | Model | Token-level Accuracy (%) | Seq.-level Accuracy (%) | Seq. Length Reduction (%) | Description of Deleted Tokens |
|---|---|---|---|---|---|
| **Simple Vowel Removal** | T5 | 100.00 | 99.93 | 0.00 | — |
| | MrT5 ($\alpha = 1e{-}4$) | 100.00 | 99.92 | 18.58 | All vowels. |
| | MrT5 ($\alpha = 1e{-}3$) | 100.00 | 99.92 | 20.15 | Start tokens and all vowels. |
| **Contextual Vowel Removal** | T5 | 100.00 | 99.91 | 0.00 | — |
| | MrT5 ($\alpha = 1e{-}3$) | 100.00 | 99.78 | 1.56 | Only start tokens. |
| | MrT5 ($\alpha = 1e{-}2$) | 99.99 | 99.72 | 18.97 | Start tokens and vowels after lowercase consonants. |
| **Sequence Merge** | T5 | 100.00 | 99.84 | 0.00 | — |
| | MrT5 ($\alpha = 1e{-}2$) | 99.99 | 99.44 | 10.15 | Start tokens and most instances of 'B'. |
| | MrT5 ($\alpha = 1.5e{-}2$) | 99.98 | 99.19 | 17.37 | All 'B's, and 'C' within 'ABC' sequences. |

## 5 CONTINUED PRE-TRAINING

In our main set of experiments, we train MrT5 models on the ByT5 span corruption task. In this pre-training objective, spans of tokens in unlabeled text data are replaced with a single *sentinel* token ID per span, and the model must fill in the missing tokens. For ByT5 and MrT5, these are spans of bytes, and the masks can potentially interfere with word boundaries.

For continued pre-training, we use the multilingual C4 (mC4) corpus (Raffel et al., 2020; Xue et al., 2021). We train all MrT5 and baseline models on 15 typologically diverse languages: English, French, Spanish, German, Greek, Bulgarian, Russian, Turkish, Arabic, Vietnamese, Thai, Chinese, Hindi, Swahili, and Urdu. Finally, we examine how each model's performance correlates with sequence length reduction rates, both overall and for each language. Following related work (Yu et al., 2023; Pagnoni et al., 2024), we compare models using bits-per-byte (BPB), a tokenizer-agnostic alternative to perplexity defined as $\text{BPB} = \mathcal{L}_{\text{CE}} \times \log_2(e)$.

**Models.** We train several MrT5 models with different PI controller target deletion ratios: $\delta \in \{0.3, 0.4, 0.5, 0.6, 0.7\}$. This is a continued pre-training setup; we load the pre-trained weights and use the architecture settings of ByT5 Small (300M parameters) and only randomly initialize the weights of MrT5's gating mechanism. Based on a sweep of different layers, we place the gating mechanism after encoder layer $l = 3$ (see Section 7). In addition to the MrT5 models, we also train several baselines:

1. **ByT5 baseline**: A ByT5 Small architecture with $\text{softmax}_1$, but without deletion.[2]

2. **Random baseline**: We implement and train a set of models with a random gating mechanism, where the choice of how the tokens are deleted is random; some number of gate values are set to $k$, and the rest are set to 0. In our experiments, we train five random models with different average deletion rates: 30%, 40%, 50%, 60%, and 70%.

3. **Fixed baseline**: We implement and train a set of models that delete the ends of words. We train five models that delete different percentages of the ends of words: 30%, 40% 50%, 60%, and 70%. For details on the implementation of this baseline, see Appendix F.1.

4. **Boundary Predictor (BP) baseline**: We implement and train a set of models that integrate an unsupervised boundary predictor module, inspired by the Hourglass Transformer architecture (Nawrot et al., 2022; 2023), which detects segment boundaries for mean-pooling representations. To generate models with different compression rates, we adjust the prior probability of a boundary, $p$, across different training runs: $p \in \{0.3, 0.4, 0.5, 0.6, 0.7\}$. Further details on the implementation of this baseline can be found in Appendix F.2.

5. **Convolutional Pooling (CP) baseline**: We implement and train a set of models that use a strided 1D convolutional layer to downsample the sequence, similar to CANINE (Clark et al., 2022). To generate models with different compression rates, we vary the stride $r$ of the convolutional layer: $r \in \{2, 3, 4\}$. For a sequence of length $N$, there will be $\frac{N}{r}$ downsampled representations.

---

[2]The validation loss for ByT5 with $\text{softmax}_1$ (0.7919) and the loss for an unaltered ByT5 (0.7917) were comparable, further motivating our use of $\text{softmax}_1$. Training settings are described in Appendix E.2.

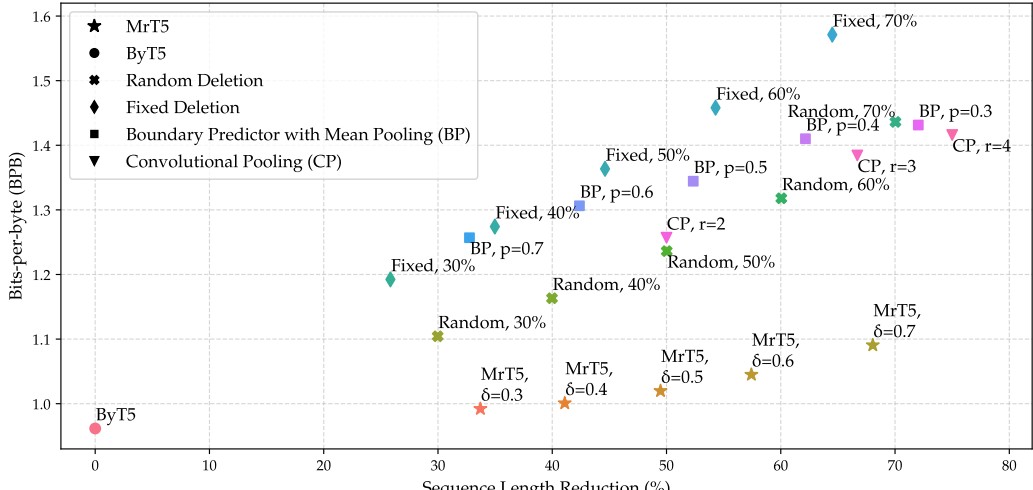

Figure 2: Span corruption BPB vs. sequence length reduction for each MrT5 and baseline model. MrT5 models consistently have much lower BPB than the baselines, and are generally competitive with unmodified ByT5, even where they achieve very large sequence length reductions.

Table 3: Average inference runtime for a single sequence (in milliseconds) for ByT5 and each MrT5 model. Percentage decrease in runtime relative to ByT5 is displayed in parentheses.

| **Model** | ByT5 | MrT5 | | | | |
|---|---|---|---|---|---|---|
| | | $\delta = 0.3$ | $\delta = 0.4$ | $\delta = 0.5$ | $\delta = 0.6$ | $\delta = 0.7$ |
| **Runtime** (ms) | 44.29 | 34.99 ($\downarrow$ 20.98%) | 31.95 ($\downarrow$ 27.85%) | 29.99 ($\downarrow$ 32.27%) | 28.24 ($\downarrow$ 36.23%) | 24.48 ($\downarrow$ 44.72%) |

For the random and fixed baselines, the gating mechanism is placed at layer $l = 3$, like the MrT5 models. For the BP and CP baselines, we place the corresponding downsampling modules at layer $l = 3$ as well. All models use $\mathrm{softmax}_1$ in their attention mechanisms. The ByT5 baseline serves as a lower bound on the best possible span corruption BPB, since it does not reduce the sequence length. For further details on model architectures, train/test dataset preparation, and optimization, see Appendix E.2.

**BPB vs. Sequence Length Reduction.** First, we compare span corruption BPB across MrT5 and baseline models with varying sequence length reduction rates. Figure 2 illustrates the relationship between BPB and sequence length reduction for each model. These results are aggregated across all languages by averaging the results per language; for language-specific plots, see Figure 6 in Appendix G. As expected, ByT5 achieves the lowest BPB but does not reduce the sequence length. While MrT5 models generally have higher BPB than ByT5, they consistently outperform other baselines at comparable sequence length reduction rates. For example, MrT5 with $\delta = 0.5$ reduces the sequence by about 50%, similar to the 50% random baseline and CP baseline with $r = 2$, yet achieves the lowest BPB. This trend holds across all MrT5 models, demonstrating the effectiveness of its gating mechanism in balancing sequence length reduction and BPB minimization.

**Runtime Speedup.** MrT5 models with a higher sequence reduction rate also have a faster inference runtime, as shown in Table 3. In particular, MrT5 models that reduce the sequence length by 50% or more can achieve 30% speedup or greater, with our implementation. These results show that hard deletion improves the efficiency of MrT5 when compared to ByT5, and this speedup can be tuned.

While our main experiments use 300M-parameter models, we also trained MrT5 at a larger 1.23B-parameter model size and observed even greater efficiency gains over ByT5, with a 44.6% reduction in runtime at a 49.5% decrease in sequence length. Additionally, the gap in BPB between MrT5 and ByT5 diminished at this scale, indicating that MrT5's deletion mechanism becomes even more effective in larger models. For the details of these larger-model experiments, see Appendix H.

**Per-language Evaluation.** We analyze how a single MrT5 model performs across 15 training languages. Figure 3 plots test set BPB against sequence length reduction for MrT5 (with $\delta = 0.5$) for each language, alongside ByT5's BPB as a baseline. The results show that MrT5 adapts its deletion rates to each language, with only a minor increase in BPB compared to ByT5. More than half of the languages exhibit a sequence reduction rate above 50%. In languages with more information-dense scripts, such as Chinese, MrT5 still performs substantial deletions but at a lower rate than for less dense scripts. These findings highlight MrT5's ability to learn and apply language-specific, context-aware deletions when trained on multilingual data.

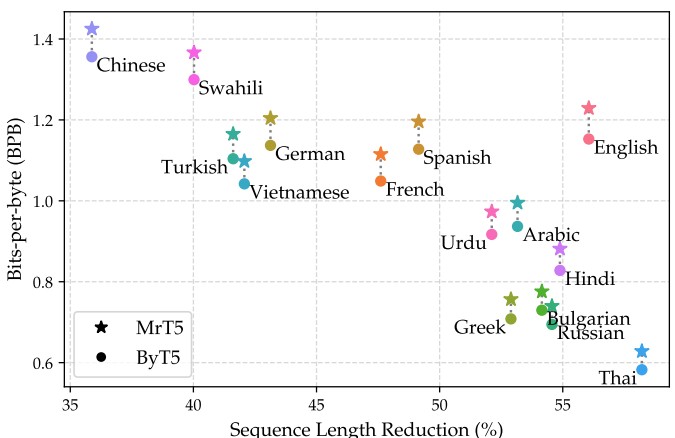

Figure 3: Average test set BPB vs. sequence length reduction for MrT5 ($\delta = 0.5$) across each of the 15 languages. ByT5 is shown for BPB comparison only (it does not reduce the sequence length). MrT5 learns language-specific sequence length reduction rates and achieves over 50% reduction in many languages with minimal effect on the BPB.

## 6 Downstream Task Evaluations

We next assess MrT5's performance on downstream tasks, specifically, two cross-lingual benchmarks (XNLI and TyDi QA) and two character-level tasks (Spelling Correction and Word Search). The two multilingual tasks evaluate MrT5's ability to understand high-level semantics and retrieval in a multilingual setting, while the character-level tasks test whether it retains its sensitivity to character-level manipulations.

**Cross-lingual Benchmarks.** We first test the cross-lingual capabilities of MrT5 using the Cross-lingual Natural Language Inference (XNLI) corpus (Conneau et al., 2018), a benchmark for cross-lingual sentence classification with 5,000 parallel examples in 15 languages. This is a cross-lingual zero-shot transfer task; models are fine-tuned on the English MultiNLI corpus (Williams et al., 2018) only, and are tested on multiple languages. The test set contains the same 15 languages our models are trained on. Our second cross-lingual evaluation employs the TyDi QA Gold Passage Task (TyDiQA-GoldP, Clark et al., 2020). In this task, given a passage that is guaranteed to contain the answer, the model must generate the correct answer to the question. TyDiQA-GoldP covers nine typologically diverse languages and comprises 204K question-answer pairs: Arabic, Bengali, English, Finnish, Indonesian, Korean, Russian, Swahili, and Telugu.

For each task, we fine-tune four models: the baseline ByT5 model, MrT5 with $\delta = 0.5$, a BP baseline with $p = 0.5$, and a CP baseline with $r = 2$. Each model is fine-tuned on top of its corresponding model from our previous continued pre-training experiment, ensuring that MrT5 and the BP/CP baselines maintain the same target compression rate established during pre-training (enforced by $\delta$, $p$, or $r$, depending on the model). This allows the models to preserve their sequence length reductions and efficiency improvements while adapting to downstream tasks. Additional training details can be found in Appendix E.3.

Table 4 presents the results on both XNLI and TyDiQA-GoldP. For the XNLI task, MrT5 is the superior model; it is the most efficient in terms of both sequence length reduction and runtime speedup (55.60% and 45.13%, respectively), and it achieves the highest task accuracy (65.31%), outperforming ByT5. The XNLI task illustrates the efficiency of MrT5's fast encoder; a 50% deletion rate results in substantial gains in runtime. This improvement can be attributed to the setup of the XNLI task, which requires large input sequences (up to 1,024 tokens) and short decoder sequences (a single token for classification). On the TyDiQA-GoldP task, MrT5 once again proves to be the

Table 4: Evaluation metrics for XNLI and TyDiQA-GoldP. XNLI task performance is measured by accuracy (chance: 33%), and TyDiQA-GoldP by EM/F1 scores. MrT5 achieves shorter sequences and faster runtimes than ByT5 while maintaining similar performance. BP and CP fall short in both efficiency and task performance. See Table 8 and Table 9 in Appendix I for per-language metrics.

| Language | Task Performance | | | | Runtime Decrease (%) | | | Seq. Len. Reduction (%) | | |
|---|---|---|---|---|---|---|---|---|---|---|
| | ByT5 | MrT5 | BP | CP | MrT5 | BP | CP | MrT5 | BP | CP |
| XNLI | 64.72 | 65.31 | 59.37 | 49.94 | 45.13 | 33.22 | 36.32 | 55.60 | 53.95 | 50.07 |
| TyDiQA-GoldP | 69.90/79.58 | 68.27/77.73 | 40.59/54.04 | 54.99/65.85 | 33.31 | 13.38 | 31.97 | 47.48 | 22.55 | 50.05 |

Table 5: Evaluation metrics for character-level tasks. MrT5 provides the best efficiency improvements (runtime and sequence length reduction), while maintaining competitive accuracy with ByT5. See Table 10 and Table 11 in Appendix I for metrics on individual test splits for each task.

| Task | Seq.-Level Accuracy (%) | | | | Runtime Decrease (%) | | | Seq. Len. Reduction (%) | | |
|---|---|---|---|---|---|---|---|---|---|---|
| | ByT5 | MrT5 | BP | CP | MrT5 | BP | CP | MrT5 | BP | CP |
| Spelling Correction | 66.73 | 63.18 | 54.81 | 57.29 | 22.30 | 16.48 | 23.13 | 50.54 | 49.30 | 50.00 |
| Word Search | 76.61 | 72.02 | 74.33 | 68.23 | 59.29 | 44.18 | 53.29 | 74.53 | 69.78 | 75.00 |

most efficient model, outperforming the alternative baselines in terms of runtime savings (33.31%). Moreover, MrT5 comes closest to matching ByT5's exact match (EM) and F1 scores, with 68.27 EM and 77.73 F1, demonstrating that it retains much of the original model's accuracy while significantly improving efficiency.

**Character-level Tasks.** We fine-tune and evaluate MrT5 and baseline models on the *Spelling Correction with Context* and *Word Search* character-level tasks from Huang et al. (2023). In the Spelling Correction task, the input is a sentence containing a spelling error, and the goal is to generate the same sentence with the error corrected. In the Word Search task, the input follows the format `definition: letters`, and the objective is to identify the substring in `letters` that, when reversed, matches the given `definition`. We selected these two tasks from the suite because they both require understanding meaning and context and involve processing longer sequence lengths.

We follow the same fine-tuning approach used for the multilingual benchmarks, using models initialized from our previous continued pre-training experiment. For Spelling Correction, we fine-tune MrT5 with $\delta = 0.5$, BP with $p = 0.5$, and CP with $r = 2$, maintaining the same target compression rates from pre-training. However, for Word Search, we adjust the compression parameters to further optimize efficiency, using MrT5 with $\delta = 0.7$, a BP baseline with $p = 0.3$, and a CP baseline with $r = 4$. We evaluate on all test splits from Huang et al. (2023), which are designed to rigorously assess a model's ability to integrate meaning and context in its predictions. Further training details can be found in Appendix E.3.

Table 5 displays the test set results for each character-level task. MrT5 stands out as the most efficient model, achieving notable reductions in both sequence length (50.54% for spelling correction and 74.53% for word search) and runtime (22.30% and 59.29%, respectively). Despite these optimizations, it maintains competitive sequence-level accuracy (63.18% for spelling correction and 72.02% for word search), making it a well-balanced choice between speed and performance. These results show that MrT5's method of sequence merging effectively preserves its sensitivity to character-level information.

## 7 ANALYSIS

**Per-sample Sequence Length Reduction.** We present a per-sample analysis of bits-per-byte and sequence length reduction for several MrT5 models and random baselines with different sequence length reduction rates. We take a sample of 1,000 English sentences from the mC4 test set and calculate the percent increase in BPB on a per-sample basis, using ByT5's BPB as the baseline (i.e. the percent increase between MrT5's BPB and ByT5's BPB for individual samples). For each sample, we also get the sequence length reduction. Across five MrT5 models with different deletion rates, we found a very weak/negligible correlation between the percent increase in BPB and the sequence

length reduction (average correlation of $r = 0.103$).[3] This reflects what we would expect from the MrT5 models; for an individual sample, MrT5 learns when it can delete more tokens without incurring a large increase in the loss. In contrast, for five random models with different sequence length reduction rates, we found a moderate positive correlation (average correlation of $r = 0.295$). When the random model removes more tokens, it is more likely to degrade performance. These results further support the observation that the MrT5 models more strategically and contextually delete tokens compared to the baselines. See Figure 7 in Appendix J for plots presenting the correlations for each individual MrT5 and random baseline model.

**Gate Placement.** We present an analysis of MrT5 models with delete gates placed at various encoder layers. To maximize efficiency, it is ideal to position the delete gate at the earliest possible encoder layer. However, placing the gate too early reduces the contextual information in the token representations, leading to more significant performance degradation compared to ByT5. We trained several MrT5 models with a target deletion ratio of $\delta = 0.5$, all using the same training setup and architecture as described in Section 5. The only variable was the placement of the delete gate, as shown in Figure 4. Inference runtime is lower and test set BPB is higher when the gate is placed in early layers. Since our goal is to place the gate as early as possible, we selected layer 3 as the optimal point for gate placement, since its BPB is comparable to layers 4 and 5. This analysis provides further evidence that MrT5 merges information into fewer tokens, since deletion at earlier, less contextual layers results in a higher BPB.

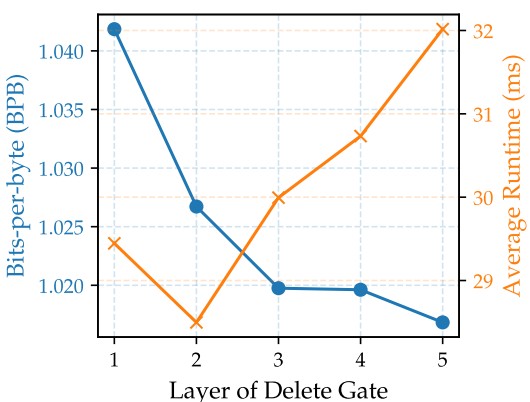

Figure 4: BPB and inference runtime for a single sequence for MrT5 models with delete gates at different layers ($l \in [1, 5]$). All MrT5 models are trained with a PI controller with a target deletion ratio of $\delta = 0.5$. BPB is consistently higher in early layers; since the gate should be placed as early as possible, we select layer 3 as optimal.

The gate placement sets an upper bound on the compute savings achievable with our model. We provide a detailed analysis of MrT5's theoretical compute savings in Appendix C. With typical hyperparameters of our tasks, we can achieve a maximum speedup of around three times with reasonable deletion rates.

## 8 CONCLUSION

In this paper, we introduce **MrT5** (**Mer**ge**T5**), a variant of the ByT5 architecture designed to address the inefficiencies of byte-level language modeling. MrT5's token deletion mechanism forces the model to merge input tokens into a more compact sequence, allowing for computational savings while preserving model performance. Our diagnostic experiments demonstrate that MrT5 effectively merges relevant context into a shorter sequence using strategies that align with task-specific objectives. In our continued pre-training experiments, MrT5 outperforms alternative deletion baselines and downsampling/pooling methods, and with multilingual training, MrT5 achieves language-specific sequence length reduction rates with minimal impact on performance. Our model learns very fast: the continued pre-training requires only a few thousand additional training steps. Furthermore, MrT5 maintains competitive accuracy with ByT5 on downstream multilingual benchmarks and character-level tasks while improving inference runtimes. This demonstrates MrT5's capacity to handle tasks requiring semantic information, while still effectively processing character-level details—the main advantage of byte-level modeling. Our work takes a significant step toward the viability of byte-level language models and eliminating the need for subword tokenization.

---

[3]When averaging correlation coefficients, we apply Fisher's Z transformation to stabilize the variance.

## ACKNOWLEDGMENTS

The authors would like to thank the members of the Stanford NLP Group and Jurafsky Lab for helpful feedback and discussions at various stages of this project. We would also like to thank Sabah Kallini for suggesting the name "MrT5." Julie Kallini is supported by a National Science Foundation Graduate Research Fellowship under grant number DGE-2146755. This work is supported in part by a grant from the Laboratory Directed Research and Development program at Sandia National Laboratories. Sandia National Laboratories is a multimission laboratory managed and operated by National Technology and Engineering Solutions of Sandia LLC, a wholly owned subsidiary of Honeywell International Inc. for the U.S. Department of Energy's National Nuclear Security Administration under contract DE-NA0003525.

## REPRODUCIBILITY STATEMENT

Steps for reproducing each of our experiments are detailed in Appendix E. Descriptions of the model architectures and training configurations/hyperparameters for our diagnostic task experiments are provided in Appendix E.1; details of the model architectures, span corruption data preprocessing steps, and training configurations/hyperparameters for continued pre-training are provided in Appendix E.2; and training configurations/hyperparameters for fine-tuning on the multilingual and character-level downstream tasks are provided in Appendix E.3. We provide our source code at https://github.com/jkallini/mrt5.

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

## A    FURTHER RELATED WORK

**Early Exit Models.** Our architecture is related to early-exit methods proposed for autoregressive Transformers (Elbayad et al., 2020; Schuster et al., 2022) and BERT (Xin et al., 2020; 2021). In contrast to previous approaches, our method is fully differentiable and does not require special training considerations or calculating the entropy of the final classifier, and the deletion decisions are made in a single layer, making the model efficient and easy to use.

**Long-context Models.** Other work has addressed long-context modeling more broadly. Perceiver AR (Hawthorne et al., 2022) uses cross-attention to map inputs to a smaller set of latent vectors, enabling autoregressive modeling over sequences up to 100K tokens. However, it was not evaluated on character-level tasks or language tasks without tokenization. Other solutions include Nugget (Qin & Van Durme, 2023; Qin et al., 2023), which encodes the whole sentence, but passes only a dynamic subset of the embeddings to the decoder. This approach does not save compute on the encoder side. Recently, GemFilter (Shi et al., 2024) has improved inference runtimes in decoder-only models by using early LLM layers as a filter to select input tokens to be processed by the full model; while this method requires no additional training, it requires two forward passes during inference.

**Additional Tokenization-free Models.** Before the Transformer, character-level LSTMs, with their compact vocabularies, were effective across a range of NLP tasks, particularly in morphologically rich languages (Gillick et al., 2016; Kim et al., 2016; Ballesteros et al., 2015; Cherry et al., 2018). Building on this idea, CharBERT (Ma et al., 2020) enhances subword models with character-level embeddings from a bidirectional GRU and a noise-aware pretraining task to improve robustness. More recently, MambaByte (Wang et al., 2024) extends the Mamba architecture (Gu & Dao, 2023) to byte-level inputs, offering a fully tokenization-free alternative. Tokenization-free approaches have also been explored in other modalities, including speech (Irie et al., 2017) and text rendered as images (Rust et al., 2023).

## B    GUMBEL-SIGMOID

The Gumbel-Sigmoid is a special case of Gumbel-Softmax (Jang et al., 2017; Maddison et al., 2017) for binary choices. Please refer to Csordás et al. (2021) for more details. Given the logit $l \in \mathbb{R}$ and random samples $u_1, u_2 \sim \mathrm{Uniform}(0, 1)$, the Gumbel-Sigmoid can be calculated as

$$s = \sigma \left( l - \log \left( \log u_1 / \log u_2 \right) \right) \tag{8}$$

where $\sigma(x) = \frac{1}{1+e^{-x}}$ is the standard logistic sigmoid and $-\log \left( \log u_1 / \log u_2 \right)$ is the Gumbel noise. We use this soft formulation of the Gumbel-Sigmoid without discretization.

## C    THEORETICAL COMPUTE SAVINGS

Here we analyze the theoretical amount of compute used by MrT5 given a deletion rate. Let's call the average length of the input sequence $N_E$ and the average length of the output sequence $N_D$. The width of the residual is $d_{\mathrm{model}}$, the dimension of the up-projection in the MLP is $d_{\mathrm{ff}}$, the encoder has $L_E$ layers and the decoder has $L_D$ layers. The deletion occurs after the $l_{\mathbf{G}}$ layer, and the average proportion of deleted tokens is $\delta$. We assume that the total size of the head projections is equal to $d_{\mathrm{model}}$, as typical for Transformers ($d_{\mathrm{head}} * N_{\mathrm{heads}} = d_{\mathrm{model}}$). Then, we can approximate the total number of multiply-accumulate operations (MACs) for the model as follows. Before deletion, the self-attention in the encoder uses $N_E d_{\mathrm{model}}^2$ MACs for both the $\mathbf{Q}$, $\mathbf{K}$, $\mathbf{V}$ and the output projections and $N_E^2 d_{\mathrm{model}}$ MACs for both the $\mathbf{A} = \mathbf{Q}\mathbf{K}^\top$ and $\mathbf{A}\mathbf{V}$ projections. The MLP layer uses $N_E d_{\mathrm{model}} d_{\mathrm{ff}}$ additional MACs. Thus, the total number of MACs used per layer is $4N_E d_{\mathrm{model}}^2 + 2N_E^2 d_{\mathrm{model}} + N_E d_{\mathrm{model}} d_{\mathrm{ff}}$. This much compute is used for the first $l_{\mathbf{G}}$ layers, after which the sequence length is reduced to $N_E(1 - \delta)$ for the remaining $L_E - l_{\mathbf{G}}$ layers. Thus, the encoder uses

$$\begin{aligned} N_{\mathrm{MACs}}^{\mathrm{encoder}} = (L_E - l_{\mathbf{G}}) \, (1 - \delta) \left( 4N_E d_{\mathrm{model}}^2 + 2N_E^2 d_{\mathrm{model}} (1 - \delta) + N_E d_{\mathrm{model}} d_{\mathrm{ff}} \right) + \\ l_{\mathbf{G}} \left( 4N_E d_{\mathrm{model}}^2 + 2N_E^2 d_{\mathrm{model}} + N_E d_{\mathrm{model}} d_{\mathrm{ff}} \right) \end{aligned} \tag{9}$$

The MACs used by the decoder can be calculated similarly, but additionally the cross attention has to be taken into account. The cross attention uses $N_E(1-\delta)d_{\mathrm{model}}^2$ MACs for the $\mathbf{K}$ and $\mathbf{V}$ projections,

$N_D d_{\text{model}}^2$ MACs for the $\mathbf{Q}$ and output projections, and $(1-\delta)N_E N_D d_{\text{model}}$ MACs for the attention matrix itself.

$$N_{\text{MACs}}^{\text{decoder}} = L_D \left(4N_D d_{\text{model}}^2 + 2N_D^2 d_{\text{model}} + N_D d_{\text{model}} d_{\text{ff}} + 2N_E(1-\delta)d_{\text{model}}^2 + \right.$$
$$\left. 2N_D d_{\text{model}}^2 + 2(1-\delta)N_E N_D d_{\text{model}}\right) \tag{10}$$

Note that $\delta = 0$ corresponds to the baseline ByT5. For MrT5 Small, $d_{\text{model}} = 1472$, $d_{\text{ff}} = 3584$, $L_E = 12$, $L_D = 4$, $l_{\mathbf{G}} = 3$, $N_E = 1024$, and $N_D = 189$. Given these numbers, we plot the total reduction in compute (compared to ByT5) as a function of $\delta$ in Fig. 5.

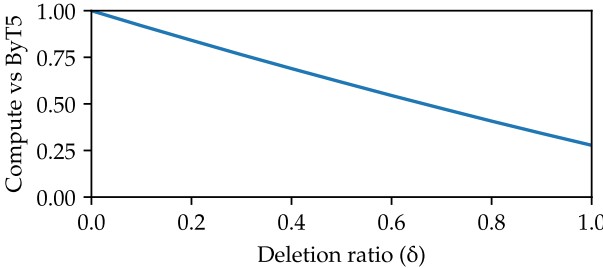

Figure 5: Reduction in the total amount of compute as a function of the deletion ratio $\delta$.

## D   REGULARIZATION OF ATTENTION SCORES

Directly adding the delete gate values to the attention scores as in Equation (2) can lead to an unintended inflation of the attention scores. This inflation would arise when training MrT5 from scratch, counteracting the intended masking effect of the delete gate values and allowing deleted tokens to influence the encoder and cross-attentions. To mitigate this issue, we introduce a regularization term designed to prevent the attention scores from inflating excessively, thereby preserving the delete gate's intended function.

Let $E$ and $C$ denote the number of encoder layers and cross-attention layers, respectively. We define $l_{\mathbf{G}} < E$ as the encoder layer where the delete gate is placed in MrT5.

For any encoder layer $e \in \{l_{\mathbf{G}}, \ldots, E\}$, let $\mathbf{Q}_e$ and $\mathbf{K}_e$ denote the query and key matrices, respectively. Similarly, for any cross-attention layer $c \in \{1, \ldots, C\}$, let $\mathbf{Q}_c$ and $\mathbf{K}_c$ denote the corresponding query and key matrices. The attention scores for these layers are computed as:

$$\mathbf{S}_e = \mathbf{Q}_e \mathbf{K}_e^\top, \quad \forall e \in \{l_{\mathbf{G}}, \ldots, E\} \tag{11}$$

$$\mathbf{S}_c = \mathbf{Q}_c \mathbf{K}_c^\top, \quad \forall c \in \{1, \ldots, C\} \tag{12}$$

To control attention score inflation, we first apply a clamping operation at a minimum threshold $m$, compute the mean of the clamped scores, and subtract $m$:

$$\mu(\mathbf{S}) = \left(\frac{1}{|\mathbf{S}|} \sum_{s \in \mathbf{S}} \max(s, m)\right) - m \tag{13}$$

where $s$ is an element of the scores matrix $\mathbf{S}$, and $|\mathbf{S}|$ represents the number of elements in the matrix. Finally, we compute the aggregate regularization loss across all encoder and cross-attention layers, excluding encoder layers before $l_{\mathbf{G}}$:

$$\mathcal{L}_{\mathbf{S}} = \frac{1}{E - (l_{\mathbf{G}} - 1) + C} \left[\sum_{e=l_{\mathbf{G}}}^{E} \mu(\mathbf{S}_e) + \sum_{c=1}^{C} \mu(\mathbf{S}_c)\right] \tag{14}$$

This final result $\mathcal{L}_{\mathbf{S}}$ represents an aggregate measure over all encoder and cross-attention layers. We use a minimum threshold $m$ is to avoid over-penalizing attention scores, which could negatively impact model performance. Instead of regularizing all attention scores, we only aim to regularize

excessively large values. We found that with $m = 5.0$, we could mitigate attention score inflation when training from scratch, and in continued pre-training experiments, the regularizer had no effect on MrT5 with no deletion ($\delta = 0.0$) compared to the baseline ByT5 model. This shows that the regularization can target excessively large attention scores, without affecting the keys and queries in a way that deteriorates the performance of the model.

The final loss is calculated as $\mathcal{L} = \mathcal{L}_{\text{CE}} + \alpha\mathcal{L}_{\text{G}} + \beta\mathcal{L}_{\text{S}}$, where $\mathcal{L}_{\text{CE}}$ is the cross entropy loss, and $\mathcal{L}_{\text{G}}$ is the delete gate loss defined in Equation (3). We can vary the strength of the regularization by tuning the parameter $\beta$. We found $\beta = 5.0$ to work well in practice, and this is the value we apply in all experiments.

# E    EXPERIMENTAL DETAILS

## E.1    SIMULATION DETAILS

**Model Architectures.** We train our diagnostic models with 3 encoder layers and 3 decoder layers, and we use $d_{\text{ff}} = 1024$, $d_{\text{model}} = 512$, and 4 attention heads in each layer. We use $\text{softmax}_1$ for all T5 and MrT5 models. Other architectural settings match the standard ByT5 Small, resulting in an architecture with 15M parameters (5% of ByT5 Small's parameter count). All models use $\text{softmax}_1$ in their attention mechanisms.

**Optimization.** We use a batch size of 128 examples and a sequence length of 64 tokens, and we train each model for a total of 30,000 gradient steps. We use the AdamW optimizer with a learning rate that linearly warms up to $1e-4$ over 3,000 steps and linearly decays. For MrT5 models, the delete gate's regularizer is enabled at 10,000 steps. We set a constant regularizer $\alpha$ throughout training and use attention score regularization, as described in Appendix D.

## E.2    CONTINUED PRE-TRAINING DETAILS

**Model Architectures.** All MrT5 and baseline models use the model configuration of a standard ByT5 Small, which has $d_{\text{ff}} = 3584$, $d_{\text{model}} = 1472$, 12 encoder layers, 4 decoder layers, 6 attention heads in each layer, and 300M total parameters. MrT5's gating mechanism introduces an additional 2,945 parameters; the boundary predictor baseline's downsampling module introduces an additional 5M parameters; and the convolutional pooling baseline's downsampling module introduces and additional 6.5M parameters. All models use $\text{softmax}_1$ in their attention mechanisms.

**Data.** When training on the span corruption objective, we calculate the corrupted spans such that the average masked span length is 20 tokens with a noise density of 15%—that is, 15% of tokens in the sequence are masked out—following the specification outlined in the ByT5 paper. To avoid training models for multiple epochs, we ensure that the samples drawn from the mC4 corpus are sufficiently large. Additionally, we extract equal-sized samples for each language (in terms of bytes) from the mC4 training split.

For evaluation, we sample each language's test set from its mC4 validation split. Each language is tested on 10,000 examples, except for Swahili and Urdu, which only have 2,800 and 9,300 examples in their validation splits, respectively. We sample a *disjoint* sample of 16,000 examples of English C4 to use as a validation set during training.

**Optimization.** We train each model for 5,000 gradient steps over batches of $2^{20}$ tokens (i.e. an encoder sequence length of 1024 with an effective batch size of 1024). We use the AdamW optimizer with an initial learning rate of $1e-4$ with linear decay and no warmup. For MrT5, we use a controller to optimize different deletion ratios and use attention score regularization, as described in Appendix D.

At test time, we use an eval batch size of $2^{14}$ tokens (i.e. an encoder sequence length of 1024 with a batch size of 16). We use the last model checkpoint at step 5,000 for all evaluations.

## E.3    DOWNSTREAM TASK DETAILS

**Models.** All MrT5 and baseline models are initialized from a corresponding model in our continued pre-training experiments and inherit the configuration of a standard ByT5 Small. This configuration

includes $d_{\text{ff}} = 3584$, $d_{\text{model}} = 1472$, 12 encoder layers, 4 decoder layers, 6 attention heads per layer, and a total of 300M parameters. All models employ $\text{softmax}_1$ in their attention mechanisms.

The specific model selected for initialization is based on the desired compression rate for fine-tuning. For instance, since we aim for 50% compression in XNLI, we initialize fine-tuning from the MrT5 model pre-trained with $\delta = 0.5$, and we continue to fine-tune with that target deletion ratio.

**Data.** For XNLI, all models are trained on the English MNLI training set and evaluated on the full XNLI test set in English, as well as zero-shot in 14 additional languages. For TyDiQA-GoldP, the multilingual training set is split into 80% for training and 20% for validation, and evaluation is performed on the separate TyDiQA-GoldP test set. For the character-level tasks, we use the provided train/validation sets for training and evaluate on each task's respective test split.

**Optimization.** For all downstream tasks, we fine-tune all MrT5 and baseline models using the AdamW optimizer with a cosine learning rate schedule and no warmup. MrT5 models use a PI controller to achieve a target deletion ratio and use attention score regularization, as described in Appendix D. Training and evaluation details for each task are summarized in Table 6.

Table 6: Fine-tuning and evaluation details for the XNLI, TyDiQA-GoldP, Spelling Correction, and Word Search downstream tasks. The maximum sequence length shown is for the encoder. LR denotes the initial learning rate used during fine-tuning.

| Task | Steps | Epochs | Batch Size | Max Seq. Length | LR | PI Controller |
|---|---|---|---|---|---|---|
| XNLI | 6,000 | $\approx 7.82$ | 512 | 1,024 | 1e−4 | $\delta = 0.5, k_p = 0.5, k_i = 1\text{e}{-}5$ |
| TyDiQA-GoldP | 6,000 | $\approx 35.07$ | 512 | 2,048 | 5e−5 | $\delta = 0.5, k_p = 0.5, k_i = 1\text{e}{-}5$ |
| Spelling Correction | 207,690 | 35 | 16 | 64 | 3e−5 | $\delta = 0.5, k_p = 0.5, k_i = 1\text{e}{-}5$ |
| Word Search | 319,670 | 65 | 16 | 128 | 5e−4 | $\delta = 0.7, k_p = 1\text{e}{-}3, k_i = 1\text{e}{-}6$ |

For XNLI, we use an evaluation batch size of 16; for the character-level tasks, we use an evaluation batch size of 64; and for TyDiQA-GoldP, we use an eval batch size of 1 (since the answers are generated by the model for each sequence). For each task, the eval maximum sequence length matches the training sequence length. The final checkpoint is used for evaluation. For XNLI and TyDiQA-GoldP, this is step 6,000; for Spelling Correction, this is step 200,000; and for Word Search, this is step 310,000.

# F  Baseline Implementation Details

## F.1  Description of Fixed Deletion Baselines

The fixed deletion baseline deletes tokens at layer $l = 3$ using deterministic rules based on the token identity. All tokens/columns corresponding to whitespace, punctuation, and symbolic characters are identified: `\t`, `\n`, `␣`, `!`, `"`, `#`, `$`, `%`, `&`, `'`, `(`, `)`, `,`, `+`, `,,`, `-`, `.`, `/`, `:`, `;`, `<`, `=`, `>`, `?`, `@`, `[`, `\`, `]`, `;`, `_`, `;`, `{`, `|`, `}`, `˜`, ``. These separator tokens are used to locate word boundaries. Then, based on the fixed deletion percentage, the delete gate will drop the tokens/columns corresponding to the ends of words. For example, if the target percentage is 50%, the delete gate will remove the tokens corresponding to the final two characters of a five-letter word, and the final three characters of a six-letter word.

## F.2  Description of Boundary Predictor Baselines

As an additional baseline, we implement a version of T5 that downsamples a sequence using a *Gumbel-Sigmoid boundary predictor*, enabling end-to-end learnable segmentation, similar to Hourglass Transformers (Nawrot et al., 2023). This method allows adaptive sequence compression by mean-pooling adjacent tokens into variable-length segments, which are subsequently processed at a reduced resolution. We first summarize the boundary prediction and downsampling approach of Nawrot et al. (2023), followed by a description of the modifications we made to adapt the approach to our setting.

**Boundary Prediction via Gumbel-Sigmoid.** The Hourglass Transformer employs a boundary predictor to learn segment boundaries in an autoregressive fashion. The goal is to find a sequence of

segment boundaries $\mathbf{b} \in \{0, 1\}^N$, where $N$ is the input sequence length. Consider two tokens $x_t$ and $x_{t+1}$, where $t \in [1, N]$ is the index of the token $x_t$ in the sequence. Given the hidden state $\mathbf{h}_t$ of $x_t$, a multi-layer perceptron with parameters $\phi$ produces the probability of a boundary $\hat{b}_t$ between token $x_t$ and token $x_{t+1}$:

$$\hat{b}_t = p(b_t = 1) = \sigma(\text{MLP}_\phi(\mathbf{h}_t)). \tag{15}$$

To enable gradient-based optimization while sampling discrete boundary indicators, the boundary predictor uses the *Gumbel-Sigmoid trick* with a relaxed Bernoulli distribution:

$$\bar{b}_t = \sigma \left[ \log \left( \frac{\hat{b}_t u}{(1 - \hat{b}_t)(1 - u)} \right)^{1/\tau} \right], \quad u \sim \text{Uniform}(0, 1) \tag{16}$$

$$b_t = \mathbf{1}_{\bar{b}_t > 0.5} \tag{17}$$

where $\mathbf{1}_x$ is the indicator function. Following Nawrot et al. (2023), we use the straight-through estimator (Hinton, 2012; Bengio et al., 2013) to make this operation differentiable:

$$b_t = \mathbf{1}_{\bar{b}_t > 0.5} - [\hat{b}_t]_{\text{stop}} + \hat{b}_t \tag{18}$$

where $[\cdot]_{\text{stop}}$ is an operator for preventing backward gradient flow. Here, $u$ introduces stochasticity, and $\tau$ is a temperature parameter controlling the sharpness of boundary decisions.

**Segment Pooling via Algebraic Manipulation.** Once boundaries $b_t$ are determined, tokens within the same segment are aggregated using mean pooling into a shortened sequence length $S = 1 + \sum_t b_t$. This is computed via a binary assignment matrix $\mathbf{B} \in \mathbb{R}^{N \times S}$, where $\mathbf{B}_{ij} = 1$ if token $i$ is assigned to segment $j$, and 0 otherwise. The pooled segment representations $\mathbf{S} \in \mathbb{R}^{S \times d_{\text{model}}}$ (where $d_{\text{model}}$ is the model dimensionality) are computed via weighted averaging of the hidden states $\mathbf{H} \in \mathbb{R}^{N \times d_{\text{model}}}$:

$$\mathbf{S} = (\mathbf{B}^\top \mathbf{H}) \oslash (\mathbf{B}^\top \mathbf{1}_{N \times d_{\text{model}}}) \tag{19}$$

where $\oslash$ denotes element-wise (Hadamard) division and $\mathbf{1}_{N \times d_{\text{model}}} \in \mathbb{R}^{N \times d_{\text{model}}}$ is a matrix of ones. The shortened sequence $\mathbf{S}$ is then processed by subsequent Transformer layers.

**Regularization via a Binomial Prior.** To prevent trivial segmentations (i.e. excessively fine or coarse partitions), a *Binomial prior* is imposed on the expected number of segment boundaries:

$$\text{Binomial}(k; N, p) = \binom{N}{k} p^k (1 - p)^{l-k}, \tag{20}$$

where $k = \sum_t b_t$ is the total number of boundaries, $N$ is the sequence length, and $p$ is a tunable prior controlling the expected segmentation rate.

**Adaptations to the T5 Setting.** The boundary predictor method used in Hourglass Transformers required several modifications to be applicable in our experimental setting, as it was originally designed for autoregressive architectures. To enable its use with an encoder model, we introduced the following adaptations:

1. **Removal of null-group representations:** In the original method, the last segment of the sequence is removed, and a *null-group representation* is prepended to sequences to prevent potential data leakage from the future to the past in an autoregressive setting. However, in an encoder model, such representations are unnecessary and were found to degrade performance. We therefore excluded them.

2. **Handling padded sequences:** In an encoder model, sequences of varying lengths are often batched together with padding tokens. To prevent padding tokens from being included in pooling, we identify the position $i$ of the first pad token in each sequence and enforce a boundary at position $i - 1$ (i.e. setting $b_{i-1} = 1$). Additionally, any boundaries predicted beyond this position are zeroed out. This ensures that the final downsampled segment includes only non-padding tokens, preventing pad tokens from being pooled into the shortened representations.

3. **Updating the attention mask:** Token pooling reduces the sequence length dynamically, and each sequence within a batch might have different numbers of segment boundaries. To ensure that attention operates only over valid pooled representations, we construct a new attention

mask that aligns with the shortened sequence. Specifically, for each sequence in the batch, any positions beyond the number of shortened segments $S$ are masked, preventing the model from attending to extraneous representations.

4. **Handling relative position biases:** The original boundary predictor does not support models that incorporate relative position biases, which are used in the attention mechanisms of each T5 layer. To address this, we designed a solution that uses the position bias of the first token within a span as the position bias of the entire shortened segment.

In our experiments, we place the boundary predictor module at encoder layer $l = 3$, the same layer at which MrT5's delete gate is placed. Subsequent encoder layers operate on the mean-pooled hidden states $\mathbf{S}$. Following Nawrot et al. (2023), we set the Gumbel temperature to $\tau = 0.5$ for all experiments. We vary the prior probability of a boundary $p$ across training runs to obtain models with different compression rates.

## G    PER-LANGUAGE SPAN CORRUPTION EVALUATIONS

Figure 6 contains the span corruption BPB across several MrT5 and baseline models with varying sequence length reduction rates. Evaluations are performed in each of the 15 training languages individually. The shapes and colors correspond to the same models evaluated in Figure 2.

## H    CONTINUED PRE-TRAINING EXPERIMENTS WITH LARGE MODELS

We present additional continued pre-training experiments using larger model sizes. Specifically, we train one ByT5 model and one MrT5 model, both initialized from the pre-trained 1.23-billion parameter ByT5 Large. While ByT5 Small consists of 12 encoder layers and 4 decoder layers, ByT5 Large has a much deeper architecture with 36 encoder layers and 12 decoder layers. ByT5 Large also has $d_{\text{ff}} = 3840$, $d_{\text{model}} = 1536$, and 16 attention heads in each layer. MrT5 Large's gating mechanism introduces an additional 3,073 parameters. In MrT5, the delete gate is applied at layer $l = 3$, and all other data and training configurations remain consistent with those used in our primary continued pre-training experiments, as detailed in Appendix E.2. We employ a PI controller to target a deletion ratio of $\delta = 0.5$. We evaluate on a smaller batch size of 4 examples ($2^{12}$ tokens) using a test set containing 2500 examples in each of the 15 languages.

The results in Table 7 underscore the effectiveness of MrT5 at a larger model scale. Notably, the gap in bits-per-byte (BPB) between ByT5 and MrT5 is significantly reduced compared to smaller-scale models, suggesting that MrT5's performance greatly improves as model capacity increases. At the same time, the efficiency gains become even more pronounced; a 49.5% decrease in sequence length results in a 44.6% reduction in inference runtime. This improved efficiency can be attributed to the deeper encoder architecture, where only 3 out of MrT5's 36 encoder layers process the full sequence, significantly reducing computational overhead. These findings suggest that MrT5's deletion mechanism scales effectively with larger models, offering an appealing trade-off between efficiency and modeling performance in high-resource settings.

Table 7: Performance comparison of ByT5 Large and MrT5 Large ($\delta = 0.5$).

| Bits-per-byte (BPB) | | Average Runtime (ms) | | MrT5 Runtime Decrease (%) | MrT5 Seq. Len. Reduction (%) |
|---|---|---|---|---|---|
| ByT5 | MrT5 | ByT5 | MrT5 | | |
| 0.70 | 0.74 | 177.27 | 98.21 | 44.6 | 49.5 |

## I    ADDITIONAL DOWNSTREAM TASK EVALUATIONS

Table 8 contains XNLI evaluation metrics for MrT5 and all baseline models for each of the 15 XNLI languages. Table 9 contains the TyDiQA-GoldP evaluation metrics for all MrT5 and baseline models for each of the 9 TyDi QA languages. Table 10 contains evaluations on all test splits for the Spelling Correction with Context task. Table 11 contains evaluations on all test splits for the Word Search task.

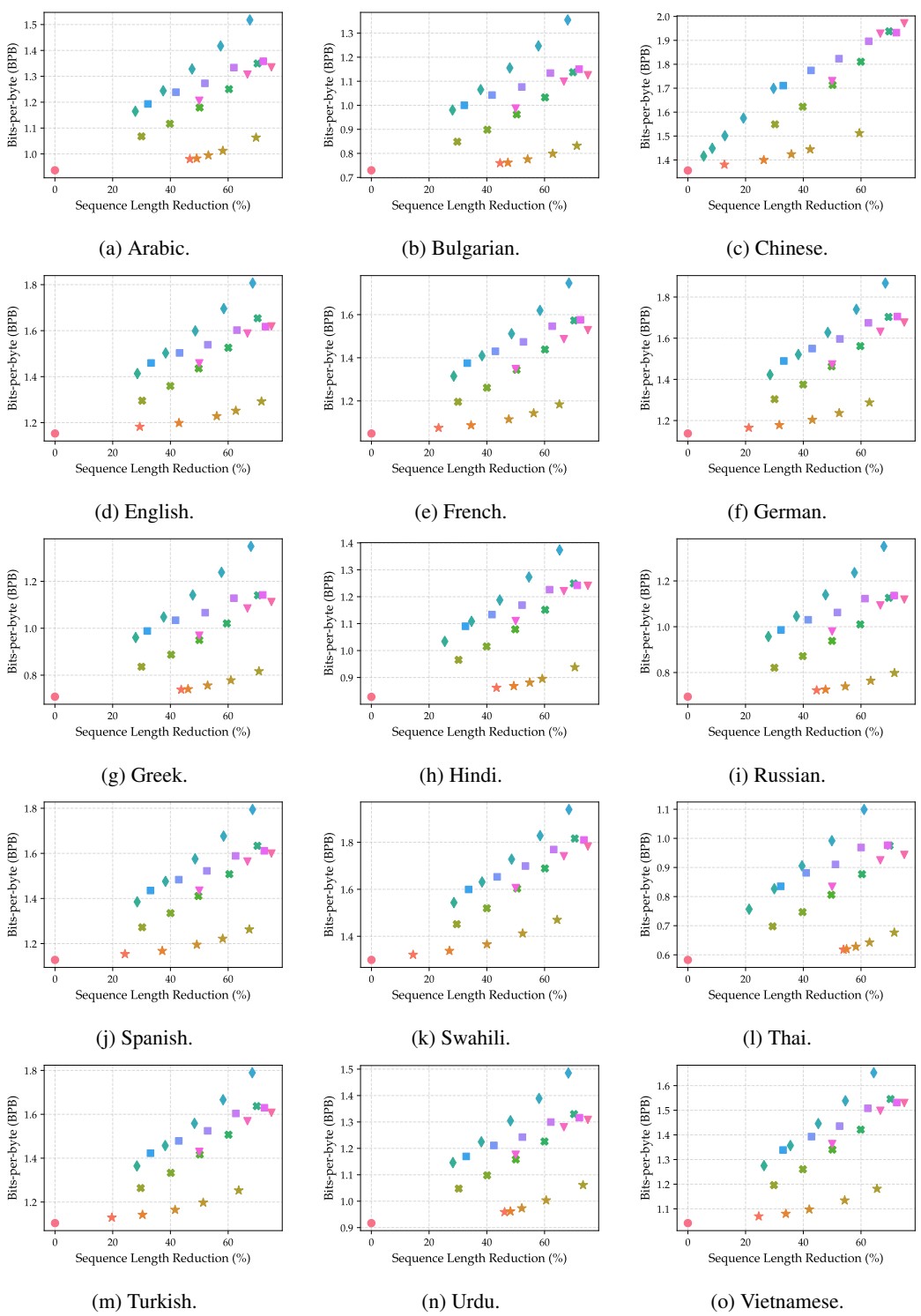

Figure 6: BPB vs. sequence length reduction for each MrT5 and baseline model, evaluated on span corruption test sets for 15 languages individually. Legend: ★ = MrT5; ● = ByT5; ✖ = random deletion baseline; ♦ = fixed deletion baseline; ■ = boundary predictor (BP) baseline with mean pooling; ▼ = convolutional pooling (BP) baseline.

Table 8: Per-language XNLI evaluation metrics for ByT5, MrT5, BP, and CP models. Except for English, all evaluations are zero-shot.

| Language | Accuracy (%) | | | | Average Runtime (ms) | | | | Runtime Decrease (%) | | | Seq. Len. Reduction (%) | | |
|---|---|---|---|---|---|---|---|---|---|---|---|---|---|---|
| | ByT5 | MrT5 | BP | CP | ByT5 | MrT5 | BP | CP | MrT5 | BP | CP | MrT5 | BP | CP |
| English | 80.30 | 80.20 | 78.90 | 73.53 | 9.01 | 5.62 | 7.33 | 5.90 | 37.64 | 18.63 | 34.50 | 50.22 | 43.38 | 50.10 |
| French | 73.93 | 73.03 | 69.02 | 55.19 | 10.76 | 6.67 | 8.21 | 7.02 | 38.04 | 23.69 | 34.75 | 50.21 | 48.50 | 50.08 |
| Spanish | 74.85 | 74.25 | 69.78 | 59.78 | 10.15 | 6.15 | 7.79 | 6.63 | 39.44 | 23.29 | 34.63 | 51.88 | 48.11 | 50.09 |
| German | 69.58 | 69.70 | 63.61 | 50.66 | 10.44 | 6.77 | 8.01 | 6.82 | 35.12 | 23.28 | 34.66 | 47.04 | 47.97 | 50.08 |
| Greek | 64.73 | 65.03 | 57.33 | 46.53 | 18.81 | 9.32 | 10.75 | 11.79 | 50.48 | 42.85 | 37.32 | 63.65 | 65.51 | 50.05 |
| Bulgarian | 67.47 | 68.14 | 61.88 | 52.79 | 17.37 | 8.63 | 10.78 | 10.96 | 50.30 | 37.96 | 36.89 | 57.98 | 54.38 | 50.05 |
| Russian | 64.21 | 66.25 | 60.48 | 51.46 | 17.80 | 8.72 | 10.85 | 11.20 | 50.99 | 39.03 | 37.07 | 64.32 | 61.27 | 50.06 |
| Turkish | 61.68 | 63.11 | 58.88 | 47.31 | 9.77 | 6.22 | 8.13 | 6.40 | 36.37 | 16.85 | 34.49 | 48.48 | 43.46 | 50.08 |
| Arabic | 62.97 | 63.61 | 57.29 | 49.34 | 14.00 | 7.82 | 9.57 | 8.96 | 44.15 | 31.63 | 36.01 | 57.98 | 54.38 | 50.06 |
| Vietnamese | 67.37 | 66.57 | 58.64 | 47.92 | 12.54 | 7.64 | 8.75 | 8.10 | 39.10 | 30.20 | 35.41 | 51.35 | 53.83 | 50.07 |
| Thai | 55.45 | 58.02 | 46.19 | 43.55 | 24.95 | 11.87 | 12.34 | 15.42 | 52.45 | 50.57 | 38.23 | 67.08 | 74.89 | 50.04 |
| Chinese | 62.12 | 60.34 | 57.92 | 44.17 | 8.07 | 4.93 | 6.96 | 5.29 | 38.89 | 13.67 | 34.44 | 48.03 | 40.04 | 50.12 |
| Hindi | 55.41 | 57.15 | 48.22 | 41.24 | 24.19 | 11.51 | 14.48 | 14.96 | 52.43 | 40.14 | 38.17 | 66.96 | 66.49 | 50.04 |
| Swahili | 60.04 | 57.92 | 55.85 | 44.57 | 9.14 | 5.92 | 7.48 | 6.01 | 35.23 | 18.16 | 34.23 | 47.34 | 44.03 | 50.10 |
| Urdu | 50.76 | 56.31 | 46.59 | 41.08 | 15.49 | 8.83 | 10.47 | 9.85 | 43.01 | 32.39 | 36.44 | 55.87 | 56.87 | 50.06 |
| All Languages | 64.72 | 65.31 | 59.37 | 49.94 | 14.17 | 7.77 | 9.46 | 9.02 | 45.13 | 33.22 | 36.32 | 55.60 | 53.95 | 50.07 |

Table 9: Per-language TyDiQA-GoldP evaluation metrics for ByT5, MrT5, BP, and CP models. When runtime relative to ByT5 is not reduced, the percent decrease in runtime is omitted (–).

| Language | Exact Match / F1 (%) | | | | Average Runtime (ms) | | | | Runtime Decrease (%) | | | Seq. Len. Reduction (%) | | |
|---|---|---|---|---|---|---|---|---|---|---|---|---|---|---|
| | ByT5 | MrT5 | BP | CP | ByT5 | MrT5 | BP | CP | MrT5 | BP | CP | MrT5 | BP | CP |
| Russian | 65.02/75.64 | 60.59/71.41 | 31.90/43.27 | 49.63/60.66 | 45.87 | 28.44 | 36.87 | 29.60 | 38.00 | 19.62 | 35.46 | 54.70 | 30.60 | 50.03 |
| Arabic | 69.27/81.84 | 68.62/80.98 | 40.17/59.72 | 60.69/75.44 | 39.13 | 24.90 | 33.29 | 26.12 | 36.36 | 14.94 | 33.27 | 56.09 | 25.37 | 50.04 |
| Bengali | 55.75/67.22 | 56.64/66.08 | 23.01/31.41 | 38.94/51.24 | 77.87 | 38.88 | 57.42 | 47.51 | 50.06 | 26.26 | 38.99 | 66.68 | 43.21 | 50.01 |
| Telugu | 77.88/85.41 | 77.43/84.87 | 29.90/42.40 | 59.04/68.78 | 57.22 | 29.96 | 44.84 | 35.70 | 47.65 | 21.63 | 37.60 | 65.33 | 36.34 | 50.03 |
| Finnish | 69.18/78.92 | 67.90/76.97 | 45.65/58.88 | 54.22/65.76 | 26.31 | 22.28 | 25.97 | 19.71 | 15.31 | 1.30 | 25.09 | 31.19 | 13.88 | 50.05 |
| Swahili | 77.96/85.28 | 76.55/82.77 | 63.33/71.95 | 60.92/67.89 | 19.00 | 17.04 | 20.84 | 16.92 | 10.33 | – | 10.93 | 40.68 | 5.39 | 50.09 |
| Korean | 58.70/66.26 | 57.61/65.66 | 31.88/40.07 | 38.04/44.67 | 31.37 | 25.28 | 29.42 | 22.30 | 19.43 | 6.22 | 28.91 | 32.77 | 17.00 | 50.04 |
| Indonesian | 75.58/84.23 | 73.10/83.19 | 50.09/65.11 | 61.06/72.10 | 28.76 | 22.16 | 27.04 | 20.81 | 22.93 | 5.98 | 27.64 | 39.27 | 16.61 | 50.06 |
| English | 63.64/73.50 | 62.50/70.93 | 36.82/51.20 | 48.41/57.79 | 31.05 | 23.07 | 28.45 | 21.48 | 25.69 | 8.36 | 30.80 | 40.48 | 21.48 | 50.04 |
| All Languages | 69.90/79.58 | 68.27/77.73 | 40.59/54.04 | 54.99/65.85 | 37.22 | 24.83 | 32.24 | 25.32 | 33.31 | 13.38 | 31.97 | 47.48 | 22.55 | 50.05 |

Table 10: Evaluation metrics for all splits for the Spelling Correction with Context character-level task. The "Dependent" split requires incorporating context to correct the spelling error; the "Independent" split does not.

| Spelling Correction Test Split | Seq.-Level Accuracy (%) | | | | Average Runtime (ms) | | | | Runtime Decrease (%) | | | Seq. Len. Reduction (%) | | |
|---|---|---|---|---|---|---|---|---|---|---|---|---|---|---|
| | ByT5 | MrT5 | BP | CP | ByT5 | MrT5 | BP | CP | MrT5 | BP | CP | MrT5 | BP | CP |
| Dependent | 49.41 | 44.20 | 36.51 | 37.40 | 3.86 | 2.98 | 3.20 | 2.94 | 22.59 | 16.99 | 23.72 | 50.85 | 49.34 | 50.00 |
| Independent | 82.11 | 80.04 | 71.08 | 74.96 | 3.90 | 3.04 | 3.28 | 3.02 | 22.05 | 16.04 | 22.62 | 50.26 | 49.25 | 50.00 |
| All Splits | 66.73 | 63.18 | 54.81 | 57.29 | 3.88 | 3.01 | 3.24 | 2.98 | 22.30 | 16.48 | 23.13 | 50.54 | 49.30 | 50.00 |

Table 11: Evaluation metrics for all splits for the Word Search character-level task. The "OOV" split contains hidden words with mT5 tokenization not seen in the training split (this does not apply to our work, since we only train byte-level models, not subword models); the "Paraphrase" split contains definitions from *The Online Plain Text English Dictionary*, testing the ability to understand context; the "Overlap" split contains overlapping hidden words; and the "Paraphrase + Overlap" split contains both paraphrased definitions and overlapping hidden words.

| Word Search Test Split | Seq.-Level Accuracy (%) | | | | Average Runtime (ms) | | | | Runtime Decrease (%) | | | Seq. Len. Reduction (%) | | |
|---|---|---|---|---|---|---|---|---|---|---|---|---|---|---|
| | ByT5 | MrT5 | BP | CP | ByT5 | MrT5 | BP | CP | MrT5 | BP | CP | MrT5 | BP | CP |
| OOV | 78.49 | 73.96 | 78.42 | 71.25 | 6.56 | 2.63 | 3.62 | 3.02 | 59.90 | 44.79 | 53.98 | 71.77 | 69.95 | 75.00 |
| Paraphrase | 85.92 | 81.51 | 81.84 | 72.30 | 6.76 | 2.75 | 3.75 | 3.14 | 59.39 | 44.51 | 53.52 | 71.88 | 69.72 | 75.00 |
| Overlap | 77.31 | 72.72 | 77.41 | 74.86 | 6.77 | 2.76 | 3.80 | 3.18 | 59.19 | 43.92 | 53.06 | 77.69 | 69.80 | 75.00 |
| Paraphrase + Overlap | 60.37 | 55.48 | 57.01 | 51.89 | 6.76 | 2.77 | 3.80 | 3.18 | 59.06 | 43.82 | 53.00 | 75.40 | 69.78 | 75.00 |
| All Splits | 76.61 | 72.02 | 74.33 | 68.23 | 6.75 | 2.75 | 3.77 | 3.15 | 59.29 | 44.18 | 53.29 | 74.53 | 69.78 | 75.00 |

## J  ADDITIONAL PER-SAMPLE ANALYSES

Figure 7 shows per-sample correlation plots between the increase in BPB and the sequence length reduction for five MrT5 models and five random baseline models with different deletion rates. The percent increase in BPB is measured relative to ByT5's BPB for that example.

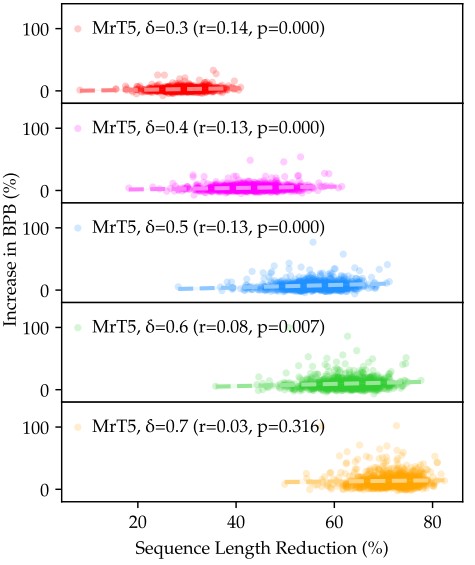 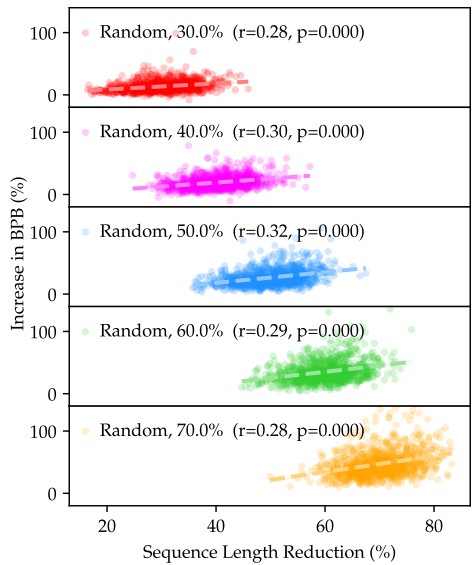

(a) MrT5 models. All models show very weak to negligible correlation between BPB increase and sequence length reduction, showing that MrT5 can delete at different percentages depending on the sequence, without incurring a significant performance drop.

(b) Random baseline models. All models show a weak to moderate positive correlation between BPB increase and sequence length reduction.

Figure 7: Percent increase in span corruption BPB vs. sequence length reduction for each (a) MrT5 model and (b) random baseline model. Percent increase is calculated using ByT5's BPB as a baseline for a particular sample. Each point represents a single sample.

