# OpenReview forum: "MrT5: Dynamic Token Merging for Efficient Byte-level Language Models"
_ICLR.cc/2025/Conference — ICLR 2025 Poster_

### Official Review · Reviewer_cEPL · 2024-10-30

**Soundness:** 2
**Presentation:** 3
**Contribution:** 2
**Rating:** 3
**Confidence:** 5

**Summary:**

This paper introduces MrT5, a ByT5 variant incorporating a learned delete gate for dynamically reducing the byte sequence length during the encoding process. By removing tokens from the sequence, the model is encouraged to implicitly merge the information from the deleted tokens into those which remain. The authors find that this reduction in sequence length leads to significant gains in runtime efficiency at inference time, with minimal differences to baseline performance. They show the deletion mechanism can reduce sequence length in zero-shot scenarios with new languages and see further improvements when training with multi-lingual data.

**Strengths:**

**Originality**
Most previous work in this area focus on algorithms for learning “what to keep” when performing sequence compression. This paper provides an interesting alternative by framing the problem as learning “what can be removed”. The proposed deletion mechanism can be added to existing architectures, making their solution easy to integrate into existing models via fine-tuning.

**Quality**
The authors conducted several levels of experimentation including simple tasks for gaining an intuition about the mechanism of their approach, a more difficult span completion task to compare against baselines, and down-stream tasks to assess cross-lingual semantic understanding and sensitivity to character-level manipulations.

**Clarity**
The paper is well-organized and clearly written. The ideas, methodology, and findings were easy to follow and understand.

**Significance**
The paper focuses on improving tokenization-free language models, which is a very relevant topic of interest to the community. Tokenization leads to several issues in LMs, including those listed by the authors as motivation: sensitivity to character-level noise (spelling errors), number representation for mathematical reasoning, and inconsistent compression rates across languages.

**Weaknesses:**

**1** The results are sensitive to the $/alpha$ hyperparameter. The gate regularization loss does not directly relate $\alpha$ to the desired compression rate, leading the authors to manually tune $\alpha$ or use the proposed P-controller to optimize for a desired compression during training. Previous work such as those cited by the authors (Nawrot et al., 2023) use a binomial loss to directly incorporate the desired compression rate into the regularization term. The same approach could be used here to remove the additional complexity.

**2** The location of the gate in a specific network also needs to be tuned. The authors show that for implicit merging of contextualized tokens the gate needs to be placed later in the network, reducing the cost-savings of the approach.
The results of the synthetic experiments were not convincing. The deletion gate seemed to only target the correct characters when sequence length was significantly reduced, resulting in a steep drop in sequence-level accuracy.

**3** There were no comparisons with other token-merging models in any of the experiments. It would have been nice to see a head-to-head comparison of this approach with the unsupervised boundary predictor from the cited (Nawrot et al., 2023) paper or the faster Toucan model from (Fleshman and Van Durme, 2023). Stronger baselines are needed throughout.

 **4** The models aren’t trained to convergence (footnote 4), and the potential 15% improvement in the baseline through continued training might not translate to MrT5, increasing the performance gap.

**Questions:**

**Q1** When training from scratch, what prevents the model from learning to set G=-30 (minimizing gating loss) and inflating QK^T for all tokens to offset the impact? It seems this would give the model additional capacity without increased loss.

**Q2** Does the specific value of k impact results when fine-tuning existing models, or does k=-30 seem to always work well?

**Q3** If the models are trained until convergence does the performance gap between MrT5 and ByT5 increase?

**Q4** Why was softmax_1 used in the ByT5 baseline? Wouldn’t the unmodified ByT5 perform better?

**Q5** The paragraph beginning at Line 180 suggests that a sequence’s compression rate depends on the least compressed sequence in the batch. Are the rates reported in the paper based on this batch-dependent compression rate or the rate you’d get with a batch size of 1?

**Q6** Similarly, are the performance metrics reported using this explanation of batch processing, or are they reported as if sequences were fully compressed?

**Q7** In Figure 3, why is the English-only trained model doing better than English on many of the other languages?

---

> ### Author Response · Authors · 2024-11-15
> **Response to Reviewer cEPL**
>
> We thank reviewer cEPL for the thoughtful comments on our work. We appreciate that the reviewer found our paper to be clearly written, well-organized, and supported by thorough experiments. We would like to address the reviewer’s concerns and questions in detail.
>
> ### Weaknesses
> 1. The reviewer sees our use of a regularizer with an $\alpha$ hyperparameter as a limitation, and cites the binomial loss of Nawrot et al. (2023) as an alternative method. The main problem is that Nawrot et al.'s regularizer requires binary decisions, which would not apply to our “soft” deletion method. Furthermore, we see our regularizer as a more flexible variation; without the controller, MrT5 can discover its own compression rate, and with a controller, MrT5 can be pushed toward a specific compression rate. It is also worth noting that the binomial loss of Nawrot et al. (2023) also includes a hyperparameter that must be tuned, and their boundary predictor is less lightweight, requiring 1M additional parameters compared to our 4k parameter gate. Our method is also effective with only a minimal amount of continued pre-training. We do not believe that their method will compete in the same continued pre-training setup.
> 2. The reviewer says that merging of contextualized tokens needs to happen later in the network. This is not the case, and we place the gate at an early layer for the most significant compute gain. For the span corruption and downstream tasks, the gates were placed at an early layer (l = 3, which is 1/4 of the ByT5 Small encoder), which resulted in deletions with minimal impact on task performance. Deleting at any reasonable layer (l >= 3) did not result in a performance impact, as shown in Figure 4. We only placed the gate at later layers for the simulations, and this was due to the simple nature of those tasks, where even a T5 model only developed non-trivial attention patterns at later layers. The synthetic experiments were designed to demonstrate that contextual deletions can be learned, rather than to optimize inference runtime at an early layer.
> 3. It is not the case that the delete gate only targets the correct characters when there is too much deletion. We show that MrT5 is capable of achieving the ground truth deletion rate as specified by the synthetic task. The only synthetic model that had a steep drop in accuracy was the MrT5 model with $\alpha$ = 1e–3 trained on the Sequence Merge task. This model was actually deleting more characters than specified for the task, resulting in a performance drop, as noted in the description given in Table 2c. These are copy tasks, so dropping characters that must be copied makes the task significantly harder. The other models that targeted relevant patterns did not see a significant performance drop, showing that MrT5 can be tuned to have the appropriate amount of deletion for a task.
> 4. We appreciate the reviewer’s suggestion to explore additional baselines. The Toucan model (Fleshman and Van Durme, 2023) speeds up the cross attention in decoding. This is negligible for improving the inference runtime of ByT5, which carries the majority of the computation in the encoder. We will be actively working on experiments adapting the boundary predictor of Nawrot et al. (2023) for encoders and will report our findings in another revision.
> 5. We fine-tune all models on MNLI for 4,000 steps, which is ~10 epochs. The ByT5 authors fine-tune the model for 262,144 steps, which would be ~650 epochs, given their setup. We find this to be excessive for MNLI, and this is what we intended to convey in footnote 4. Due to resource constraints, we do not train for as many epochs. However, none of the claims in our paper depend on this comparison to the fine-tuning in the ByT5 paper. Our focus is on speeding up ByT5.

---

> > ### Author Response · Authors · 2024-11-15
> > **Response to Reviewer cEPL (continued)**
> >
> > ### Questions
> > 1. > When training from scratch, what prevents the model from learning to set G=-30 (minimizing gating loss) and inflating QK^T for all tokens to offset the impact? It seems this would give the model additional capacity without increased loss.
> >
> > **Answer:** Our use of $\text{softmax}_1$ in the attention mechanism prevents the model from learning to set $G=-30$ (see 208-213). With $\text{softmax}_1$, setting $G=-30$ for all tokens in the sequence would cause the sum of attention scores to go to zero, emulating the effect of deleting all tokens. This reflects an impact on the loss, even with soft deletion.
> >
> > 2. > Does the specific value of $k$ impact results when fine-tuning existing models, or does $k=-30$ seem to always work well?
> >
> > **Answer:** We found that $k=-30$ worked well for all our experiments. Larger values of $k$ downweight tokens but do not emulate deletion very well (we tried $k=-10$), causing discrepancies between hard and soft deletion performance. We did not observe discrepancies between hard and soft deletion when using $k = -30$ (see the table provided in the response to Reviewer 91Fx above).
> >
> > 3. > If the models are trained until convergence does the performance gap between MrT5 and ByT5 increase?
> >
> > **Answer:** See (5) above.
> >
> > 4. > Why was $\text{softmax}_1$ used in the ByT5 baseline? Wouldn’t the unmodified ByT5 perform better?
> >
> > **Answer:** We employ $\text{softmax}_1$ consistently across all models to ensure that any performance variations are not attributed to $\text{softmax}_1$, but rather to other components of the architecture. Additionally, we evaluated the impact of $\text{softmax}_1$ on the ByT5 baseline. Our results showed that the loss for ByT5 with $\text{softmax}_1$ (0.7805) was nearly identical to the loss for the unaltered ByT5 (0.7815), further reinforcing our decision to use $\text{softmax}_1$ across all baselines.
> >
> > 5. > The paragraph beginning at Line 180 suggests that a sequence’s compression rate depends on the least compressed sequence in the batch. Are the rates reported in the paper based on this batch-dependent compression rate or the rate you’d get with a batch size of 1?
> >
> > **Answer:** The compression rates reported in the paper are batch-independent (i.e. the rate with a batch size of 1). However, any runtime speed-ups that we report (such as Table 3) are batch-dependent using the batch sizes specified in the appendices. These results demonstrate that we can achieve significant speed-ups despite the dependence on the least compressed sequence in the batch.
> >
> > 6. > Similarly, are the performance metrics reported using this explanation of batch processing, or are they reported as if sequences were fully compressed?
> >
> > **Answer:** If by “performance” the reviewer means inference-time speed-up, these are reported on entire batches, as explained in Q5 above. If by “performance” the reviewer means task loss or accuracy, these are entirely independent of the batch size and would not change based on the batch size.
> >
> > 7. > In Figure 3, why is the English-only trained model doing better than English on many of the other languages?
> >
> > **Answer:** The “English-only” models are still trained on top of a pre-trained ByT5 Small, which was trained on multiple languages and has better CE loss on certain languages other than English.
> >
> > We are grateful to the reviewer for the thorough feedback on our work, and we hope that our response addresses your concerns and questions.

---

> > > ### Comment · Reviewer_cEPL · 2024-11-19
> > > **Thank you for response**
> > >
> > > The detailed response and clarifications are very helpful. Here are some thoughts with respect to the author responses to the questions posed:
> > >
> > > 1. This question was in respect to equation (2) Even with softmax_1, the model could seemingly learn to set G=-30 for all tokens (minimizes gating loss) and inflate the magnitude of Q and K for all tokens such that the dot product dominates the sum (essentially ignoring the impact of G during training). I would guess this doesn’t completely happen because of regularization on the magnitude of weights, preventing Q and K from getting too large.
> > >
> > > 2. Thank you for the table.
> > >
> > > 3. The resource constraints are understandable, but pointing out a 20% shortfall in performance in footnote 4 suggests more training is required for convergence (at least for ByT5). This makes it difficult for the reader to assess whether the speed-up from MrT5 is worth potential performance drops when the models are fully trained.
> > >
> > > 4. If the goal of the paper is to demonstrate that MrT5 speeds up ByT5 with minimal loss in performance, then the unmodified ByT5 would be the appropriate baseline. Your experiments comparing ByT5 with and without softmax_1 could then be used to show that softmax_1 wasn’t the cause of performance variations.
> > >
> > > 5. Perfect, thank you. This would be worth emphasizing in the paper.
> > >
> > > 6. Thank you for clarifying.
> > >
> > > 7. Thank you for clarifying.

---

> > > > ### Author Response · Authors · 2024-11-25
> > > > **Follow-up Response to Reviewer cEPL**
> > > >
> > > > We thank reviewer cEPL for the suggestion to try additional pooling baselines. We have adapted the boundary predictor with pooling method of Nawrot et al. (2023) to our setting, and we include these additional results in a thread above: https://openreview.net/forum?id=VYWBMq1L7H&noteId=snMb0yKtDM

---

### Official Review · Reviewer_91Fx · 2024-11-04

**Soundness:** 2
**Presentation:** 4
**Contribution:** 2
**Rating:** 3
**Confidence:** 4

**Summary:**

This work improves on top of ByT5 by adding a delete gate at a certain layer to remove unimportant tokens. The proposed model trains a soft deletion layer which is fully differentiable, and then uses the same layer for hard (i.e., discrete) deletion during inference. The model only requires unsupervised data for training.

The model is evaluated on monolingual and cross-lingual pretraining tasks, showing a trade-off between cross-entropy loss and sequence length reduction. On two evaluation tasks, the model is also shown to be competitive with ByT5 while reducing sequence length.

**Strengths:**

• Clear and illustrative figure

• Very well-written and easy to read

• Demonstrate that the proposed method performs competitively with ByT5 on XNLI and Spelling Correction.

• The soft and hard deletion switch can be considered novel, at least for applying to this line of work.

**Weaknesses:**

• The baselines for MrT5 are not properly constructed (Section 5). To see how well MrT5 does, one should compare it with other orthogonal methods/models (e.g., pooling) that reduce sequence length and see how much increase in x-entropy loss they incur.

• For downstream tasks we should compare with non-byte-level models to see the gap: how far are we in terms of accuracy? What's the run-time comparison after this token deletion optimization? These questions are left unanswered in the paper.

• Just evaluating on XNLI and Spelling Correction is not enough to claim that the model is stronger than ByT5, let alone comparing comprehensively with models equipped with traditional tokenizers.

**Questions:**

Suggestions:

• 034-035: "... via algorithms such as byte-pair encoding (Sennrich et al., 2016) or SentencePiece (Kudo & Richardson, 2018) ..."
Consider rephrasing this sentence as BPE is part of SentencePiece.

• Consider discussing the relation between this work and GemFilter (https://arxiv.org/pdf/2409.17422) as both pertain to token deletion.

• The proposed method seems readily transferrable to decoder-only models, which is the most widely used architecture nowadays. Would like to see some experiments, or at least some discussions about this direction.

• 150-151: "(1) we want to avoid the overhead of executing the deletion algorithm multiple times;"
This motivation is better discussed from the perspective of "trade-off". If executing the algorithm multiple times can reduce the number of tokens/positions to process in later layers even more without compromising generation quality, then there is no reason to not do it.

• The regularizer loss only seems to encourage the increase of the number of deleted tokens, but does not encourage the gate output to converge to the extreme values (i.e., the min or max gate value). This could make hard deletion during inference less effective because merely setting a threshold may delete some "somewhat useful" tokens. Please refer to a very old paper on NALU to see how they do this: https://arxiv.org/pdf/1808.00508. Preferably this work can discuss the motivation of why or why not there is no such regularization term.

• It's an interesting choice to combine experimental setup and results into one section, but I still think it's better to present them separately.

---

> ### Author Response · Authors · 2024-11-15
> **Response to Reviewer 91Fx**
>
> We thank reviewer 91Fx for the insightful comments on our work. We are happy that the reviewer found our hard/soft deletion method to be novel, and that they found the paper to be well-written. We address the reviewer’s comments and questions below.
>
> ### Weaknesses
> 1. We appreciate the reviewer’s suggestion to include orthogonal methods as baselines; we will be actively working on experiments using the boundary predictor/pooling of Nawrot et al. 2023 and will report our findings in another revision. We note that the boundary predictor of Nawrot et al. is less lightweight, requiring 1M additional parameters compared to our 4k parameter gate. Our method is also effective with only a minimal amount of continued pre-training. We do not believe that their method will compete in the same continued pre-training setup.
> 2. Our goal in this work is to enhance the efficiency of ByT5 while preserving its task performance. We do not claim to compete with or outperform subword models. Nonetheless, we provide accuracy comparisons with a subword-based T5 model on character-level tasks: under identical training settings, our model surpasses T5's performance, which achieves 47.98% accuracy on spelling correction and 72.90% on word search, as reported by Huang et al. (2023). However, we reiterate that our focus remains on optimizing ByT5’s speed rather than benchmarking against subword-level models.
> 3. We do not claim that the model is stronger than ByT5. We provide significant inference runtime improvements to ByT5 while maintaining its task performance. The runtime gains after the token deletion operation are provided in Table 3, and they are quite significant.
>
> ### Questions
> 1. >Consider discussing the relation between this work and GemFilter (https://arxiv.org/pdf/2409.17422) as both pertain to token deletion.
>
> **Answer:** We appreciate the reviewer’s reference to GemFilter, and we will add it to the related work. We would like to note that GemFilter is concurrent work, as it was made public on September 25.
>
> 2. > The proposed method seems readily transferable to decoder-only models, which is the most widely used architecture nowadays. Would like to see some experiments, or at least some discussions about this direction.
>
> **Answer:** Our approach is transferable to decoder-only models, but the deleted tokens must be re-inserted in order to perform next-token prediction—a nontrivial modification to our method. We are exploring this direction, but we found this to be out-of-scope of the current work that seeks to speed up ByT5. It is a great next step, though.
>
> 3. >150-151: "(1) we want to avoid the overhead of executing the deletion algorithm multiple times;" This motivation is better discussed from the perspective of "trade-off". If executing the algorithm multiple times can reduce the number of tokens/positions to process in later layers even more without compromising generation quality, then there is no reason to not do it.
>
> **Answer:** By “overhead” we are referring to additional time required to execute the algorithm. We agree that this can be discussed as a trade-off, and we will incorporate this in the revision.
>
> 4. > The regularizer loss only seems to encourage the increase of the number of deleted tokens, but does not encourage the gate output to converge to the extreme values (i.e., the min or max gate value). This could make hard deletion during inference less effective because merely setting a threshold may delete some "somewhat useful" tokens.
>
> **Answer:** Throughout our experiments, we thoroughly tested that loss and task metrics for hard and soft deletion matched; in other words, soft deletion during training served as a true proxy for hard deletion at inference. We found that an additional regularizer that pushes the gate to its extreme values was unnecessary, and ultimately omitted it for simplicity. Here, we provide accuracy comparisons for hard and soft deletion on our downstream tasks. The results clearly show that there is little or no difference between the two.
>
> | Task                   | Hard Deletion Accuracy | Soft Deletion Accuracy |
> |------------------------|------------------------|-------------------------|
> | XNLI (English)         | 78.88                  | 78.84                   |
> | XNLI (All Languages)   | 49.63                  | 49.65                   |
> | Spelling Correction    | 56.07                  | 56.05                   |
> | Word Search            | 74.30                  | 74.25                   |
>
> We also thank the reviewer for the presentation and phrasing suggestions, which we will incorporate in the revision. We are grateful for the reviewer’s comments, and we hope that our response addresses your concerns and questions.

---

> > ### Author Response · Authors · 2024-11-25
> > **Response to Reviewer 91Fx (Continued)**
> >
> > We thank reviewer 91Fx for the suggestion to try additional pooling baselines. We have adapted the boundary predictor with pooling method of Nawrot et al. (2023) to our setting, and we include these additional results in a thread above: https://openreview.net/forum?id=VYWBMq1L7H&noteId=snMb0yKtDM

---

### Official Review · Reviewer_uaeo · 2024-11-04

**Soundness:** 3
**Presentation:** 3
**Contribution:** 3
**Rating:** 5
**Confidence:** 4

**Summary:**

This work builds on ByT5 and introduces a new token deletion mechanism that dynamically determines how to remove unnecessary tokens from the sequence without compromising performance.

The authors incorporate additional neural layers to learn the optimal token deletion process. During training, a loss function is introduced to encourage the model to progressively delete tokens (softly) or to achieve a specific ratio. During inference, the layers identify and discard unimportant tokens in a hard manner.

Controlled experiments on synthetic tasks demonstrate that the method, MrT5, can effectively learn to compress the input sequence. Further results on downstream tasks, such as XNLI, also support the authors' claims.

**Strengths:**

* A simple yet effective method that enables models to dynamically learn how to delete tokens from byte-level inputs.
* Controlled experiments and results on downstream tasks support the authors' claims.
* MrT5 demonstrates competitive inference speed compared to ByT5.

**Weaknesses:**

* If I understand correctly, during training, the byte tokens are deleted softly, meaning there are still significant burdens for byte-level language models given that standard attention has quadratic time complexity, which limits their scalability to larger sizes.
* The experiments presented utilize moderate model sizes, which may constrain the overall persuasiveness of the proposed method.

**Questions:**

If possible, I am interested in comparing the effectiveness of additional parameters introduced by MrT5 in learning to delete tokens.

---

> ### Author Response · Authors · 2024-11-15
> **Response to Reviewer uaeo**
>
> We thank reviewer uaeo for the thoughtful comments on our paper. We appreciate that the reviewer found our method to be simple yet effective for improving the inference speed of ByT5 while maintaining its performance. Below, we address the reviewer’s comments and concerns.
>
> ### Weaknesses
> While tokens are deleted softly during training, our continued pre-training approach **requires only a small number of steps (3,000) to effectively adapt the existing ByT5 model**.
>
> Furthermore, models are typically **used far more for inference than for training**, so optimizing inference runtime yields the most significant gains in practical utility. In the current paradigm, the shelf-life of LLMs is very long; they are trained just once but used extensively for inference.
>
> Take ByT5 Small as an example. In the past month alone, it was [downloaded over 2 million times](https://huggingface.co/google/byt5-small). If we assume this download rate—2 million times per month—has been consistent since its release three years ago, the model would have been downloaded approximately 72 million times. However, this is likely a conservative estimate, as download rates were probably much higher during its initial release.
>
> If each download resulted in just one inference, **the model would have executed 72 million forward passes for inference—dramatically surpassing the 1 million forward passes it executed during training**, even with this conservative estimate. This highlights how optimizing its inference runtime would be much more impactful for its real-world usability.
>
> Due to resource constraints, we only performed experiments on moderate model sizes. With additional resources, we hope to scale up our approach in future work.
>
> ### Questions
> The MrT5 gating mechanism only adds 4,417 parameters to ByT5 Small, so both models have ~300M parameters. **The additional parameter count is almost negligible**, especially when compared to related approaches. For example, the boundary predictor of Nawrot et al. 2023 introduces 1M additional parameters.
>
> We would again like to thank the reviewer for their comments, and we hope that our response addresses any concerns and questions.

---

> > ### Author Response · Authors · 2024-11-25
> > **Response to Reviewer uaeo (Continued)**
> >
> > We thank reviewer uaeo for the suggestion to provide results with larger model sizes. Please see the overall comment above for these results: https://openreview.net/forum?id=VYWBMq1L7H&noteId=iSXvMIv7Ta

---

### Official Review · Reviewer_T4gu · 2024-11-04

**Soundness:** 3
**Presentation:** 3
**Contribution:** 3
**Rating:** 6
**Confidence:** 4

**Summary:**

This paper presents MrT5 (Merge-T5), a more efficient variant of ByT5 that introduces dynamic sequence shortening through a learned token delete gate after the first few encoder layers. Though it is called Merge-T5, no tokens are merged together, only deleted or preserved.
The method is a lightweight addition to a typical transformer architecture, with fully-differentiable soft deletion used during training via attention-masking and a regularizer with a tuned weight to adjust the degree of deletion, and hard deletion at inference time.
Performance is shown to be comparable to ByT5 (+/- 2% accuracy) while reducing sequences by up to 80% and runtime up to ~55%.
Only 4,000 steps are necessary to adapt a ByT5 model to a MrT5 model.

Experiments are well designed and illustrate the properties of the method in different settings.
First, synthetic tasks (vowel deletion, token merging) are used to understand learned deletions with controlled settings and small 31M parameter models.
Next, they introduce token deletion in continued pre-training of ByT5-small both on English and multilingually, ablating components of the mechanism and different degrees of deletion. The task is span corruption loss, which for byte-level input may affect character and word boundaries.
MrT5 is (generally) able to delete bytes in other scripts even if they have been unseen (if the model has been pretrained on English only), but results in slightly higher losses than ByT5. Multilingual pretraining reduces but does not remove the loss difference between MrT5 and ByT5. For Chinese, though, token deletion does not occur zero-shot.
--> It is not clear the significance of the differences in CE in Figure 2 & 3 here - it would be nice if this could be made clear.
Finally, MrT5 is evaluated on downstream tasks, XNLI and 2 character-level tasks.
For English, MrT5 outperforms ByT5 with a reduction in sequence length and inference time of ~50%, while averaged across all languages there is a slight performance degradation of ~2% (though a significant speedup).
On two English character-level tasks (contextual spelling correction and word search), MrT5 again saw ~2% accuracy drop compared to ByT5 in exchange for a 30-55% runtime decrease.
When comparing different layer placements, it seems that a middle layer (3) balances training stability and deletion level, allowing contextual representations to be learned before deleting tokens while still pruning sufficient tokens for efficiency gains.

It would be good if it were made clearer what the possible performance cost of introducing the deletion gate is, and some sense of the variability across languages, on explicit task performance as well as the loss presented in Figure 3.

**Strengths:**

A straightforward and lightweight addition to ByT5 models which can provide significant efficiency improvements (up to 80%) with a small trade-off in accuracy (~2%) compared to ByT5 (and often small improvements for English).

**Weaknesses:**

Efficiency gains (though generally significant) can come at a slight performance cost for non-English languages, and it is not clear how much variance in this cost there may be.

- Without continued training in non-English scripts, there can be performance drops for non-English languages with less efficiency improvements. Figure 3 suggests that for Chinese for example, there may be almost no reduction in seq length zero-shot for the span corruption task, and some languages have relatively large drops in CE compared to ByT5
- For the downstream experiments, the results are reported in aggregate for non-English languages in the main text. While the results are broken out by languages in the Appendix, a mention of the variance across languages and a comment on the relationship between these results and those in the previous section would make it clearer what the potential trade-offs are and when they arise

**Questions:**

Questions:
- I'd like to see the 'all languages' split for the downstream tasks discussed in the main text to match the CE analysis -> is it the case for example that you see a performance hit and no sequence length reduction for Chinese when the deletion gate is used zero-shot on a task, following Figure 3? Essentially, it would be nice to clearly state what the possible performance cost of introducing the deletion gate is, and some sense of the variability across languages, on explicit task performance as well as the loss presented in Figure 3.
  - There is not much difference in performance between MrT5 and ByT5 for Chinese XNLI in Appendix Table 6, and there is ~20% length reduction - why do you think there is a difference to Chinese in the top half of Figure 3 with only a ~2% length reduction? Similarly, for Swahili, there seemed to be a significant difference in CE in Figure 3 zero-shot, but a smaller gap in Table 6. Why do you think that is? I suggest addressing these inconsistencies in the main text and providing possible explanations for the differences between training and downstream task performance.

Suggested citations:
- [Cherry et al 2018](https://arxiv.org/abs/1808.09943) learns to delete characters for temporal compression with character-level models in MT
- [Limisiewicz et al 2024](https://arxiv.org/pdf/2403.10691) uses morphologically inspired compression for byte sequences to create MyT5, a ByT5 variant


Presentation nits:
- L99: "the main the limitations" -> "the main limitations"
- How α was set was not super clear to me. L196: "For most of our experiments, we set α by hand, which allows the model to dynamically set the deletion ratio [based on the loss]." This paragraph says α was set by hand most of the time, but that it is easier to allow α to dynamically change. Do any of the experiments in the main text allow α to change? (It doesn't seem so?) Why not, if it is easier?

---

> ### Author Response · Authors · 2024-11-15
> **Response to Reviewer T4gu**
>
> We thank reviewer T4gu for their thorough feedback on our work. We are pleased that the reviewer found our work to be clear, acknowledging that it is a lightweight addition to ByT5 that significantly improves its inference runtime. Below, we address the constructive comments provided by the reviewer.
>
> ### Weaknesses
> 1. We recognize that our model’s zero-shot performance for non-English languages has a larger drop compared to English. While perfect zero-shot transfer from English to all other languages is not expected, MrT5's deletion gate—trained on minimal English span corruption data—demonstrates impressive zero-shot transfer capabilities across several languages. Furthermore, with just a small amount of additional pre-training on multiple languages (3,000 steps), the model's cross-lingual performance improves substantially.
> 2. We agree with the reviewer's suggestion to provide a more detailed analysis of the variance across languages in the XNLI task experiments to improve our discussion of the trade-offs. We will incorporate a more in-depth discussion of the multilingual results in the main text.
>
> ### Questions
> 1. >There is not much difference in performance between MrT5 and ByT5 for Chinese XNLI in Appendix Table 6, and there is ~20% length reduction - why do you think there is a difference to Chinese in the top half of Figure 3 with only a ~2% length reduction? Similarly, for Swahili, there seemed to be a significant difference in CE in Figure 3 zero-shot, but a smaller gap in Table 6. Why do you think that is?
>
> **Answer:** The reviewer mentions great points regarding our multilingual span corruption and XNLI task results that we clarify here. For the XNLI task discussed in Section 6, we first train the MrT5 model using multilingual span corruption data. (This is the bottom model in Figure 3). After this continued pre-training, we fine-tune the model specifically on the MNLI task, which uses only English data. This approach means that the model’s gating mechanism has been trained on multilingual data, but solely within the continued pre-training phase.
>
> Next, we evaluate the model—fine-tuned on English-only MNLI—on the XNLI task, which spans 15 languages. This evaluation is therefore a zero-shot transfer *with respect to the XNLI task*, as the model has not seen these languages during fine-tuning. Our results demonstrate that effective fine-tuning on downstream tasks can be performed in English only, yielding strong zero-shot performance across other languages.
>
> 2. > I suggest addressing these inconsistencies in the main text and providing possible explanations for the differences between training and downstream task performance.
>
> **Answer:** In Section 6, we state that we use a MrT5 model that was tuned with multilingual span corruption data before fine-tuning it on English MNLI data. We will emphasize this more and include a more thorough discussion of multilingual results.
>
> 3. > How $\alpha$ was set was not super clear to me. L196: "For most of our experiments, we set $\alpha$ by hand, which allows the model to dynamically set the deletion ratio [based on the loss]." This paragraph says $\alpha$ was set by hand most of the time, but that it is easier to allow $\alpha$ to dynamically change. Do any of the experiments in the main text allow $\alpha$ to change? (It doesn't seem so?) Why not, if it is easier?
>
> **Answer:** For the downstream task experiments (XNLI and character-level tasks) we use a controller to allow $\alpha$ to dynamically change in order to target a specific deletion ratio. For the synthetic experiments and span corruption task, we set an initial $\alpha_0$ and do not let it dynamically change, which nudges the model to delete, but ultimately allows it to discover its own deletion ratio. We found this to be the appropriate approach for those exploratory experiments, to study the behavior of MrT5 without being pushed toward a specific deletion ratio.
>
> We appreciate the reviewer for the suggested citations of Cherry et al. 2018 and Limisiewicz et al. 2024, and we will include these in the related work of the revised paper.
>
> We would like to thank the reviewer again for their insightful comments on our paper. We hope that our response has addressed your concerns and questions.

---

### Author Response · Authors · 2024-11-22
**Additional Experiments with Pooling Baselines**

Reviewers 91Fx and cEPL suggested comparing MrT5 to baselines that incorporate pooling, such as the boundary predictor with pooling method proposed by Nawrot et al. (2023). In response, we implemented this pooling baseline and report its performance relative to MrT5 in this section. **We also include these new results in Appendix E of the revised paper.**

It is important to note that the boundary predictor method by Nawrot et al. required several adaptations in order to be applied in our experimental setting, as it was originally designed for decoder architectures. Specifically, we made the following modifications to enable its use with an encoder-only model:

1. **Removal of null-group representations**: Null-group representations, which are padded at the beginning of sequences, were excluded as they are unnecessary in an encoder-only context. We found that they hurt model performance.
2. **Handling relative position biases**: The original method does not support models with relative position biases, which are used in the attention mechanism of each layer. To address this limitation, we devised a solution that uses the position biases associated with the initial boundary as the position bias for the span of pooled tokens.

We integrated the boundary predictor and pooling module after the third layer, consistent with the placement of the delete gate for MrT5 in both span corruption and downstream task experiments. We follow Nawrot et al. and use a temperature of 0.5 for the boundary predictor. The results of this comparison are presented below.

**Overall, in both the continued pre-training and downstream task experiments, MrT5 outperforms Nawrot et al.’s method in terms of task loss or accuracy while achieving comparable or better compression rates. These results highlight MrT5's ability to effectively balance performance and efficiency, offering a superior alternative to the pooling-based approach.**

# Continued Pre-training

We first evaluate the cross entropy loss on the English span corruption task for MrT5 models with models trained using Nawrot et al.’s boundary predictor method. All models are trained with the same hyperparameter/training configuration and the same number of gradient steps, as described in Appendix A.2.

For MrT5, we use a controller to target specific sequence length reduction rates and train five models with target rates of 15%, 30%, 40%, 60%, and 70%. Similarly, we train models using Nawrot et al.’s method with priors of 0.85, 0.7, 0.6, 0.4, and 0.3, which correspond to the same target reduction rates of 15%, 30%, 40%, 60%, and 70%, respectively.

**For all pairs of models with similar compression ratios, MrT5 achieves much lower cross-entropy loss compared to the models using Nawrot et al.’s method.**

| Target Compression (%) | Cross Entropy Loss (MrT5) | Seq. Len. Reduction (%) (MrT5) | Cross Entropy Loss (Nawrot) |  Seq. Len. Reduction (%) (Nawrot) |
|--------------------------|-------------------------------|------------------------------------|-------------------------------|------------------------------------|
| 15                | 0.79                          | 15.71                              | 1.01                          | 18.77                              |
| 30                | 0.80                          | 30.50                              | 1.06                          | 33.64                              |
| 40                | 0.81                          | 41.28                              | 1.08                          | 44.10                              |
| 60                | 0.83                          | 60.80                              | 1.16                          | 64.11                              |
| 70                | 0.87                          | 72.24                              | 1.19                          | 73.43                              |

---

> ### Author Response · Authors · 2024-11-22
> **Additional Experiments with Pooling Baselines (Continued)**
>
> # Downstream Tasks
>
> For the downstream tasks, all models are trained with the same hyperparameter/training configurations and for the same number of epochs, as specified in Appendix A.3 of the paper. We adjust the prior for the models that use Nawrot’s boundary predictor in order to test different compression rates. For XNLI, we first train a model with multilingual span corruption data that uses Nawrot et al.’s method before fine-tuning it on XNLI, to be consistent with the MrT5 experiments.
>
> ## XNLI
>
> For XNLI, we train a model with prior=0.5 to have a comparable compression rate to MrT5. **Across all languages, MrT5 outperforms Nawrot et al.’s method.** We also note that the MrT5 model has more dynamic compression rates for different languages, adapting to the specific information density of a particular language. In Nawrot et al.'s paper, their experiments required setting different explicit priors for different languages, which is a limitation in terms of flexibility when compared to our approach.
>
> | Language     | Accuracy (%) (ByT5) | Accuracy (%) (MrT5) | Accuracy (%) (Nawrot) | Seq. Len. Reduction (MrT5) | Seq. Len. Reduction (Nawrot) |
> |--------------|--------------------------|--------------------------|--------------------------|-------------------------------|-------------------------------|
> | English      | 76.47                   | 78.84                   | 71.48                   | 52.56                         | 52.02                         |
> | French       | 53.90                   | 51.89                   | 47.17                   | 47.58                         | 52.89                         |
> | Spanish      | 55.99                   | 53.96                   | 51.17                   | 50.67                         | 53.16                         |
> | German       | 45.68                   | 45.94                   | 41.88                   | 47.61                         | 52.80                         |
> | Greek        | 52.01                   | 52.75                   | 43.79                   | 54.79                         | 55.63                         |
> | Bulgarian    | 56.31                   | 53.62                   | 46.12                   | 47.82                         | 47.81                         |
> | Russian      | 55.35                   | 55.97                   | 43.61                   | 47.95                         | 48.57                         |
> | Turkish      | 44.86                   | 42.91                   | 40.68                   | 46.72                         | 51.86                         |
> | Arabic       | 51.89                   | 47.13                   | 42.26                   | 43.78                         | 56.65                         |
> | Vietnamese   | 48.63                   | 47.51                   | 46.08                   | 46.51                         | 51.93                         |
> | Thai         | 48.35                   | 44.07                   | 40.23                   | 61.07                         | 48.54                         |
> | Chinese      | 50.70                   | 50.44                   | 40.72                   | 24.96                         | 53.76                         |
> | Hindi        | 43.87                   | 40.25                   | 39.91                   | 43.56                         | 46.30                         |
> | Swahili      | 40.64                   | 38.61                   | 42.38                   | 48.48                         | 52.99                         |
> | Urdu         | 45.48                   | 40.88                   | 38.97                   | 38.32                         | 55.78                         |
> | All Languages       | 51.34                   | 49.65                   | 45.10                   | 46.82                         | 52.05                         |

---

> ### Author Response · Authors · 2024-11-22
> **Additional Experiments with Pooling Baselines (Continued #2)**
>
> ## Spelling Correction with Context
>
> For the spelling correction task, we try two settings for Nawrot et al.’s method: one with prior = 0.5, and one with prior = 0.3. **MrT5 outperforms Nawrot et al.’s method in terms of both task accuracy and sequence length reduction rates.**
>
> | Split                     | Accuracy (%) (ByT5) | Accuracy (%) (MrT5) | Accuracy (%) (Nawrot, prior=0.5) | Accuracy (%) (Nawrot, prior=0.3) | Seq. Len. Reduction (%) (MrT5) | Seq. Len. Reduction (%) (Nawrot, prior=0.5) | Seq. Len. Reduction (%) (Nawrot, prior=0.3) |
> |---------------------------|--------------------------|--------------------------|-----------------------------|-------------------------------|-------------------------------|-------------------------------|----------------------------------|
> | Context Dependent    | 37.63                   | 35.05                   | 34.92                   | 33.21                                 | 78.77                         | 50.24                         | 70.19                           |
> | Context Independent  | 76.45                   | 74.76                   | 73.30                   | 71.29                                          | 78.98                         | 50.20                         | 70.25                           |
> | All Splits                    | 58.19                   | 56.07                   | 55.24                   | 53.37                                         | 78.88                         | 50.22                         | 70.22                           |
>
>
> ## Word Search
>
> For the word search task, we try two settings for Nawrot et al.’s method: one with prior = 0.3, and one with prior = 0.7. **Even with much lower sequence length reduction rates, Nawrot et al.’s method has very poor accuracy and is vastly outperformed by MrT5.**
>
> | Split                     | Accuracy (%) (ByT5) | Accuracy (%) (MrT5) | Accuracy (%) (Nawrot, prior=0.7) | Accuracy (%) (Nawrot, prior=0.3) | Seq. Len. Reduction (%) (MrT5) | Seq. Len. Reduction (%) (Nawrot, prior=0.7) | Seq. Len. Reduction (%) (Nawrot, prior=0.3) |
> |---------------------------|--------------------------|--------------------------|----------------------------|----------------------------|-------------------------------|---------------------------------|---------------------------------|
> | OOV                  | 78.05                   | 79.99                   | 59.70                     | 38.82                     | 72.52                         | 20.75                          | 78.37                          |
> | Paraphrase           | 83.67                   | 82.06                   | 19.16                     | 3.68                      | 72.27                         | 21.50                          | 78.32                          |
> | Overlap              | 77.58                   | 76.22                   | 74.09                     | 54.71                     | 76.14                         | 22.47                          | 75.84                          |
> | Paraphrase + Overlap   | 58.46                   | 57.60                   | 14.75                     | 3.14                      | 73.95                         | 21.53                          | 78.20                          |
> | All Splits                     | 75.37                   | 74.30                   | 38.68                     | 22.45                     | 73.91                         | 21.76                          | 77.51                          |

---

### Author Response · Authors · 2024-11-25
**Additional Experiments with Large Model Sizes**

Reviewer uaeo requested experiments with larger model sizes than we presented in the paper. Here, we show the results for continued pre-training experiments using ByT5 Large as the base model. ByT5 Large has 1.23B parameters, 36 encoder layers, 12 decoder layers, $d_\text{model} = 1536$, and $d_\text{ff} = 3840$. We train for 1500 gradient steps over batches of $2^{20}$ tokens, as this is what we could complete in the allotted time with our compute budget. All other data and optimization settings match the span corruption experiments in the paper. **We have included these new results in Appendix F of the revised paper.**

For MrT5, we place the deletion gate at layer $l=3$, and we use a P-controller to target different deletion rates. We also train a ByT5 baseline model as well as random deletion baseline models with different deletion rates.

**With a larger model size, MrT5 consistently outperforms the random baselines and achieves a loss closer to that of the ByT5 baseline than in the smaller model experiments. Furthermore, the larger MrT5 model size yields substantially greater runtime gains since more encoder layers are skipped.**

### Cross-Entropy Losses

ByT5 Loss: 0.590

| Target Reduction Rate     | 15%      | 30%      | 40%      | 50%      | 60%      | 70%      |
|-----------|----------|----------|----------|----------|----------|----------|
| MrT5      | 0.599 | 0.610  | 0.616 | 0.625 | 0.638 | 0.676 |
| RandomT5  | 0.634 | 0.679 | 0.722 | 0.763 | 0.833 | 0.966 |

### Sequence Length Reduction Rates
| Target Reduction Rate     | 15%       | 30%       | 40%       | 50%       | 60%       | 70%       |
|-----------|-----------|-----------|-----------|-----------|-----------|-----------|
| MrT5      | 20.78 | 33.35 | 42.31 | 51.13 | 59.73 | 68.04 |
| RandomT5  | 15.16 | 30.26 | 40.03 | 50.07 | 59.83 | 69.99 |

### Runtime Analysis
| Model       | Average Runtime (ms)  | Percent Decrease vs. ByT5 (%) |
|-------------|--------------|----------------------|
| **ByT5**    | 1890.15  | —                 |
| **MrT5, 15%** | 1650.61 | 12.67              |
| **MrT5, 30%** | 1388.89 | 26.52              |
| **MrT5, 40%** | 1187.09 | 37.20              |
| **MrT5, 50%** | 1058.60 | 43.99              |
| **MrT5, 60%** | 968.63  | 48.75              |
| **MrT5, 70%** | 817.66  | 56.73              |

---

### Author Response · Authors · 2024-12-04
**Overall Author Response**

As we conclude the discussion period, we again wish to highlight the comprehensive experiments conducted in response to the reviewers' feedback. These additional results have been incorporated into the revised version of our paper, and we summarize them below:

1. **Additional Experiments with Large Model Sizes.** We evaluated MrT5 alongside baseline models using a much larger model size of 1.23 billion parameters, as requested by reviewer uaeo. The larger model size proved to be better for MrT5, which not only consistently outperformed random baselines but also reduced its loss difference compared to the ByT5 baseline. Furthermore, MrT5 is much more efficient compared to ByT5 in the large model setting. This stems from the deeper encoder in the larger models—more of MrT5's encoder layers process a shortened sequence length, effectively speeding up the computation.
These results are available [here](https://openreview.net/forum?id=VYWBMq1L7H&noteId=iSXvMIv7Ta) and have been included in Appendix F of the revised paper.
2. **Additional Experiments with Pooling Baselines.** We provide a set of results with new pooling baseline models, as suggested by reviewers 91Fx and cEPL. We adapt the approach of [Nawrot et al. (2023)](https://aclanthology.org/2023.acl-long.353/) to our setting; their learned boundary predictor method was designed for decoder-only models, but we adapt it to encoders. Overall, in both the continued pre-training and downstream task experiments, MrT5 outperforms Nawrot et al.’s method in terms of task loss or accuracy while achieving comparable or better compression rates. These results highlight MrT5's ability to effectively balance performance and efficiency, offering a superior alternative to the pooling-based approach.
See this [thread](https://openreview.net/forum?id=VYWBMq1L7H&noteId=snMb0yKtDM) for the results, which have also been included in Appendix E of the revised paper.

We believe these additional experiments address the main concerns raised by the reviewers. In the responses to each of the reviewers, we have been thorough in answering all of their questions or comments. We respectfully request that the reviewers reconsider their evaluations and adjust their scores accordingly.

---

### Meta-Review · Area_Chair_auUJ · 2024-12-21

**Metareview:**

This paper introduces MrT5, an extension of the ByT5 architecture that incorporates a "delete gate" mechanism to dynamically remove unnecessary tokens from the input sequence. The gate learns to selectively discard tokens during the encoding process, leading to significant reductions in sequence length and inference time. The authors study the effectiveness of MrT5 on various tasks, including XNLI and character-level tasks, while maintaining competitive performance with ByT5.

The reviewers have praised the novelty and the efficiency of the approach and the clarity of the paper. The reviewers raised issues including limited baselines to which the authors have provided extensive comparisons. They also have questioned scope of evaluation, and limited/non-uniform improvement for non-English languages. The authors again have provided acceptable responses. While the reviewers have not raised their scores, I believe the rebuttal sufficiently addresses most of the concerns raised in the reviews and thus, I recommend acceptance. I encourage the authors to include all the new results in their revised draft.

**Additional Comments On Reviewer Discussion:**

The main points raised by the reviewers were:
1. Lack of comparison to pertinent baselines such as Nawrot et al (2023). The authors provided substantial comparisons both on tasks and on efficiency
2. Lack of detailed results in the main paper for non-English languages which the authors have promised to add.
3. Experiments with larger model sizes, which the authors have also provided.
4. Limited tasks. This remains a concern which I believe future work should look at.

---

### Decision · Program_Chairs · 2025-01-22

Accept (Poster)